# Pathogenesis and Immunomodulation of Urinary Tract Infections Caused by Uropathogenic *Escherichia coli*

**DOI:** 10.3390/microorganisms13040745

**Published:** 2025-03-26

**Authors:** J. David García-García, Laura M. Contreras-Alvarado, Ariadnna Cruz-Córdova, Rigoberto Hernández-Castro, Marcos Flores-Encarnacion, Sandra Rivera-Gutiérrez, José Arellano-Galindo, Sara A. Ochoa, Juan Xicohtencatl-Cortes

**Affiliations:** 1Posgrado en Ciencias en Biomedicina y Biotecnología Molecular, Escuela Nacional de Ciencias Biológicas, Instituto Politécnico Nacional, Mexico City 11340, Mexico; dgarcia91@outlook.es (J.D.G.-G.); 20.93lau@gmail.com (L.M.C.-A.); 2Laboratorio de Investigación en Bacteriología Intestinal, Hospital Infantil de México Federico Gómez, Mexico City 06720, Mexico; 3Departamento de Microbiología, Escuela Nacional de Ciencias Biológicas, Instituto Politécnico Nacional, Mexico City 11340, Mexico; san.rg19@gmail.com; 4Laboratorio de Investigación en Inmunoquímica, Hospital Infantil de México Federico Gómez, Mexico City 06720, Mexico; ariadnnacruz@yahoo.com.mx; 5Departamento de Ecología de Agentes Patógenos, Hospital General “Dr. Manuel Gea González”, Mexico City 14080, Mexico; rigo37@gmail.com; 6Laboratorio de Microbiología Molecular y Celular, Biomedicina, Facultad de Medicina, Benemérita Universidad Autónoma de Puebla, Puebla 72420, Mexico; mflores31@hotmail.com; 7Laboratorio de Investigación en Virología, Hospital Infantil de México Federico Gómez, Mexico City 06720, Mexico; josearellanogalindo@yahoo.com.mx

**Keywords:** uropathogenic *Escherichia coli*, colonization factor, cytokine, toxin

## Abstract

Urinary tract infections (UTIs) are a leading cause of illness in children and adults of all ages, with uropathogenic *Escherichia coli* (UPEC) being the primary agent responsible. During colonization and subsequent infection of the urinary tract (UT), UPEC requires the expression of genes associated with virulence, such as those that encode the fimbrial adhesins FimH, PapG, and CsgA, as well as the presence of the TosA protein and the flagellar appendages of the bacteria. However, for colonization and infection to be successful, UPEC must overcome the host’s immunological barriers, such as physical barriers, expressed peptides and proteins, and immune cells found in the UT. In this context, the UT functions as an integral system where these factors act to prevent the colonization of uropathogens. Significant genetic diversity exists among UPEC strains, and the clonal complex ST131 represents one of the key lineages. This lineage has a high content of virulence genes, multiple mechanisms of antibiotic resistance, and a high frequency of extended-spectrum β-lactamases (ESBLs). New knowledge regarding protein structures known as adhesins and their role in the infection process can help identify therapeutic targets and aid in the design of vaccines. These vaccines could be based on the development of chimeric fusion proteins (FimH + CsgA + PapG), which may significantly reduce the incidence of UTIs in pediatric and adult patients.

## 1. Introduction

*General characteristics.* Urinary tract infections (UTIs) are among the main causes of morbidity in all age groups worldwide [1,2]. In 2023, the Ministry of Health of Mexico reported that UTIs are the second leading cause of morbidity, with a total of 3,365,799 identified cases (https://epidemiologia.salud.gob.mx/anuario/2023/principales/nacional/grupo_edad.pdf; accessed 31 January 2025). Recent data from the “Global Burden of Disease Study 2019” revealed the incidence, mortality, and disability-adjusted life years (DALYs) associated with UTIs across 204 countries from 1990 to 2019 [3]. In addition, these studies were conducted at different sociodemographic levels and at different national, regional, sex, and age levels. An analysis of these studies showed that the absolute number of UTI cases increased by 60.40%, from 252.25 million (95% CI: 223.31–279.3) in 1990 to 404.61 million (95% CI: 359.43–446.55) in 2019 [3]. Identifying UTIs in children in a timely manner is essential for providing timely treatment. Prompt identification of UTIs in children is essential for effective treatment, particularly because cystitis and pyelonephritis can often be hard to distinguish in children under two years of age. In the United Kingdom, 80% of UTIs in children, mainly infants, can go unnoticed, owing to the presence of nonspecific signs such as fever of 38 °C or higher, irritability, vomiting, decreased appetite, lethargy [4], and improper collection of urine samples [5]. In recent years, recommendations have been made for the diagnosis, treatment, and follow-up of children with UTIs, as well as for imaging studies (renal and bladder ultrasound), to identify underlying urological anomalies and their consequences [4,5,6]. The guidelines for diagnosing UTIs in infants emphasize preventing renal scarring. Research has highlighted the long-term consequences of renal scarring, which may include reduced kidney function, hypertension, and even end-stage renal disease [7,8]. The timely diagnosis of UTIs in both children and adults should always be a priority. This approach allows for the identification of the pathogen, enabling prompt treatment and eradication. However, there are existing recommendations concerning the diagnosis and treatment of UTIs in pediatric patients. Recent studies have indicated that the urinary tract (UT) is a common source of infections in children of all ages [4]. These infections are risk factors for potential complications following UTIs, including congenital anomalies of the kidney and UT, as well as vesicointestinal dysfunction. It is crucial to consider a diagnosis of UTI in every child who presents with fever without an apparent source of infection. Several factors must be accounted for appropriate management, such as distinguishing between lower UT infections (lUTIs) and upper UTI (uUTIs), selecting the appropriate age-specific urine collection method, assessing risk factors, obtaining a positive urine culture based on the Kass–Sandford criterion, accurately identifying the uropathogen, and applying suitable treatment on the basis of clinical observations and laboratory findings. In conclusion, several important factors must be considered to effectively manage pediatric patients with UTIs [4]. In adult patients, UTIs pose a significant health issue, mainly due to therapeutic failures linked to infections caused by multidrug-resistant (MDR) and extreme drug-resistant (XDR) uropathogenic *Escherichia coli* (UPEC) strains. An MDR strain is defined as one that is resistant to at least one antibiotic from three or more different classes of antimicrobial agents. In contrast, an XDR strain is resistant to at least one antibiotic from all classes but susceptible no more than two classes of antibiotics [9]. In women, several risk factors contribute to the development of UTIs, including loss of estrogen, changes in the vaginal microbiota, and increased vaginal pH. The signs and symptoms of a UTI in adults are typically clear and are characterized by cystitis, which includes dysuria (painful urination), polyuria (increased urination), urgent urination, and suprapubic pain. In cases of pyelonephritis, additional symptoms such as fever exceeding 38 °C, lower back pain, and systemic symptoms may be present [10,11,12]. Laboratory analyses with positive urine cultures (considering the patient’s age), noninvasive urine sampling, clinical history, and evaluation are essential for an optimal diagnosis [13].

UTIs can be classified into two categories on the basis of their anatomical site: lower urinary tract infections (lUTIs) and upper urinary tract infections (uUTIs). lUTIs include conditions such as cystitis, whereas uUTIs include urethritis, pyelonephritis, and urosepsis. The uropathogens responsible for causing lUTI include *Escherichia coli*, *Staphylococcus saprophyticus*, *Staphylococcus* spp., *Proteus* spp., *Klebsiella* spp., *Enterococcus* spp., *Pseudomonas* spp., and various yeast species [14]. Notably, *E*. *coli* accounts for more than 80% of the uropathogens associated with uUTIs. Moreover, more than 80% of community-acquired UTIs and 20–40% of hospital-acquired UTIs are associated with UPEC [15]. Our working group reported that one of the main problems with UTIs in the pediatric population is the presence of MDR and XDR clinical strains of UPEC and the production of extended-spectrum beta-lactamase (ESBL) [16]. Different guidelines for the optimal management of UTIs in pediatric patients have been proposed in several studies [17]. First-line antibiotics, such as trimethoprim-sulfamethoxazole, amoxicillin-clavulanate, and cephalosporins (2nd- and 3rd-generation, cefixime, and cefotaxime), are mainly used to treat pediatric UTIs [18]. Antibiotics such as fosfomycin, nitrofurantoin, and gentamicin have also been used as alternative antibiotics for the treatment of UTIs [19].

### 1.1. UPEC Colonization of the UT

Colonization is the initial event that promotes the establishment of UPEC in the UT of the host, subsequently leading to a UTI. UPEC adheres to the bladder epithelium as part of the colonization process, allowing it to evade urine clearance, interact with the “umbrella cells” of the urinary epithelium, and, in some cases, undergo an intracellular invasion process [20]. UPEC can internalize and multiply within host cells during this intracellular invasion, helping it evade the immune response. This ability diminishes the effectiveness of antibiotics and promotes their persistence by forming intracellular bacterial communities (IBCs) and quiescent intracellular reservoirs (QIRs) [21]. IBCs play several critical roles in UTIs: (i) they help bacteria persist in UTIs; (ii) they protect bacteria from antibiotics and antimicrobial treatments; (iii) they shield bacteria from the host’s immune response; (iv) they promote cell exfoliation, facilitating exposure of the transitional epithelium; and (v) their reactivation enables UPEC to exit through filamentation, initiating a new infectious cycle. This process leads to rUTIs and supports bacterial persistence and reactivation once the triggering stimulus has ceased [15]. Our working group characterized a collection of 178 clinical strains of urinary *E. coli* (UEc) [22] and 129 clinical strains of UPEC belonging to serotype O25b (EcO25b) from children with UTIs. The UEc and EcO25b strains presented a high prevalence of genes that encode various fimbrial adhesins, such as *fimH*, *papG*, *ecpA*, and *csgA* [23]. These adhesins are associated mainly with the adhesion process, mediating the intimate interaction between bacteria and host cells [24]. Additionally, each adhesin recognizes specific receptors in the host urothelium: (i) the FimH adhesin recognizes mannosylated residues, (ii) PapG adhesin variants recognize globosides (GbO1 to GbO5) in the renal epithelium, and (iii) the CsgA adhesin binds to fibronectin and fibrinogen [24].

### 1.2. Fimbrial and Nonfimbrial Adhesins in UPEC

Fimbria types 1 and P participate in the uropathogenesis of *E*. *coli* via the Chaperone-Usher (CU) assembly mechanism [24,25]. In the CU system, chaperones are the proteins responsible for preventing the polymerization of structural proteins in the periplasm, keeping them soluble until their recognition by the accommodator protein, which catalyzes the correct folding of the pilins [25].

#### 1.2.1. Type 1 Fimbriae

The expression of type 1 fimbriae has been linked to biofilm formation, adherence to and invasion of the uroepithelium, yeast agglutination, and immune response evasion by invading macrophages [26]. Type 1 fimbria contribute to the pathogenicity of *E*. *coli*, which is involved in the colonization of various niches from the intestine, genitals, and UT to other organs [27]. Type 1 fimbriae are encoded by the *fimAICDFGH* operon, which includes the regulatory genes *fimB*, *fimE*, and *fimX* and the *fimS* switch region. The FimA protein is a pilin that forms the fimbrial main subunit (filament), FimC is a periplasmic chaperone, and FimD is an accommodator protein. The FimI protein is a homolog of FimA; the function of FimI is unknown, but FimI is essential for the expression of fimbrial proteins [28].

The tip of the fimbria comprises the docking proteins FimF and FimG, which are involved in binding a single copy of the FimH adhesin [29]. The FimH adhesin protein contains two domains: the pilin domain, which interacts with the FimG protein, forming a flexible junction, and the lectin domain, which forms a binding pocket for a receptor that recognizes mannosylated residues [29]. The FimH adhesin exhibits genetic variability within the *fimH* gene, leading to the identification of several polymorphisms that give rise to different versions of the FimH protein. These polymorphisms can induce functional changes, impacting the ability of bacteria to adhere to various surfaces and tissues within their host [30,31]. A study involving *E*. *coli* strains collected from the intestinal mucosa of children with inflammatory bowel disease revealed that the amino acid sequence of the FimH protein is modified. These data suggest that commensal strains can adapt to changes in their microenvironment [32].

Although the exact number of polymorphisms in the *fimH* genes has not been determined, the high frequency of structural mutations in the fimbrial adhesins of extraintestinal pathogenic *E. coli* (ExPEC) indicates significant diversity in their sequences. This genetic variability of FimH enhances the adaptability of *E. coli* to different ecological niches and conditions within the host, thereby increasing its potential to cause infections [33]. An important polymorphism in UPEC/ST131 clones is FimH30. This polymorphism features an R166H mutation that weakens the interactions between the FimH domains. As a result, this change promotes stronger interactions with mannose and allows for the formation of high-affinity relaxed conformations. Compared with other variants, the expression of the FimH30 polymorphism in isogenic *E. coli* clones of the ST131 lineage enhances adherence and invasion to human cells and facilitates the formation of highly structured biofilms [34]. The expression of type 1 fimbrial proteins is regulated by two site-specific recombinases, FimB and FimE, which promote the inversion of the *fimS* switch region in response to stress factors that positively or negatively stimulate the expression of this fimbrial protein [35]. Receptors for the fimbrial adhesin FimH, which include mannosylated uroplakins (UPK1) and the α3β1 integrin, are rich in mannosylated residues and are widely distributed in the umbrella cells of the bladder epithelium. When FimH interacts with its receptors, it promotes the internalization of UPEC by reorganizing actin through the activation of RHO family GTPases [14]. Once inside, UPEC can form intracellular bacterial communities (IBCs) and quiescent intracellular reservoirs (QIRs), which are precursors to bacterial persistence in the UT [14]. Our working group described the frequency of type 1 fimbriae in UPEC strains from complicated UTIs (cUTIs) [92.1% (164/178)] and of clone ST131/O25b [95.2% (120/126)] in pediatric patients [22,23]. Our findings align with previous observations in commensal *E*. *coli* strains and other clinical samples, mainly due to the high conservation of type 1 fimbriae in this microorganism [36].

#### 1.2.2. P Fimbriae

The distal part of pyelonephritis-associated fimbriae is anchored to the PapG adhesin, which interacts with glycosphingolipids consisting of Gal-α-(1,4) Gal residues. Three alleles have been described: PapGI, PapGII, and PapGIII, which bind to GbO_3_, GbO_4_ (both abundant in human uroepithelial cells), and GbO_5_ (abundant in canine cells but not in human cells) globoside variants, respectively [21]. PapG variations in clinical strains of UPEC are an adaptation mechanism of bacteria that allows high adaptability to colonize and cause infection in humans efficiently [37]. Recent studies have shown a strong association between the PapGII variant in clinical UPEC strains belonging to the B2 phylogenetic group and several medical conditions. Phylogroup B2 is recognized as a pathogenic group associated with UTIs and characterized by genes encoding various virulence factors, including adhesin genes, toxins, and iron acquisition systems [38,39,40]. Strains belonging to this phylogroup are also known for their ability to form biofilms [41], which increases their pathogenicity and increases the likelihood of severe infections. Furthermore, UPEC strains from phylogroup B2 have shown an increasing trend in the acquisition of multidrug resistance [42]. This variant is related to the development of pyelonephritis in adult women and children, acute prostatitis in men, and cases of bacteremia [43,44,45,46]. Additionally, the PapGIII variant has been linked to the occurrence of cystitis in women, men, and children [47,48]. In contrast, the PapGI variant has a low prevalence in humans with various clinical syndromes [49].

P fimbriae are encoded by the *papBAHCDJKEFG* operon and are clustered in the pathogenicity island (PAI) PAI-CFT073-*pheV* (PAI ICFT073). P fimbria transcription is regulated by the transcriptional repressor PapI and the transcriptional activator PapB [50]. The PapA pilin protein and the PapH protein form the major and minor subunits, respectively. PapC is the accommodator protein, PapD is the chaperone, and PapG is the adhesin [50]. Recently, the prevalence of *pap* alleles was reported to be 34.3% (61/178) for *papG*II and 1.7% (3/178) for *papG*III in a collection of UEc strains [22]. Interestingly, a higher prevalence was observed in the EcO25b collection, associated with ST131 and phylogroup B2; 80.9% (102/126) amplified the *papG*II gene, and 2.4% (3/126) amplified the *papG*III gene. In contrast, the *papG*I allele has not been identified in pediatric clinical strains of UPEC [23].

#### 1.2.3. Curli Fimbriae

Amyloid structures such as curli fimbriae use an extracellular assembly mechanism called the nucleation/precipitation (NP) pathway, also known as the type VIII secretion system. A wide variety of microorganisms produce functional amyloid components as structural support that favors the integrity of biofilms, a mechanism that promotes the colonization of abiotic and biotic surfaces [51]. UPEC biofilm is essential for the persistence and recurrence of UTIs. Biofilms protect bacteria from harsh conditions, antimicrobial agents, and antibiotics, and they also safeguard bacteria from the host immune response [51]. These biofilms can easily form on biological surfaces and medical devices, such as urinary catheters and the uroepithelium [52]. Curli are amyloid fimbria present in most strains of *E. coli*, and their expression promotes the formation of biofilms in both pathogenic and commensal strains. In addition, curli-dependent biofilms can play a significant role in UTIsassociated with medical devices, such as catheters and urinary probes [53,54]. While many pathogenic, opportunistic, and commensal bacteria of the *Enterobacteriaceae* family members produce curli, our research group has demonstrated that curli fimbria functions as an accessory protein that enhances UT colonization in a C57BL/6 mouse model. These results indicate that infection with a curli-producing strain can cause more significant damage to the bladder and kidneys than the damage caused by the same strain in which the *csgA* gene is deleted [55]. Although curli fimbria are directly involved in the formation of biofilms by UPEC strains, type 1 fimbria-dependent biofilms have also been reported on abiotic surfaces [56]. These biofilms contribute to the persistence of catheter-associated UTIs (CAUTIs) [57] and are crucial factors in the development of UTIs [58]. Furthermore, type 1 fimbria facilitate colonization of the bladder epithelium and IBCs, contributing to persistent and recurrent UTI infections. Curli fimbriae are primarily composed of CsgA proteins, which have a fine structure with aggregation and adherence properties [59]. The genes encoding the proteins involved in curli biogenesis are located in the *csgBAC* operon, which is transcribed divergently. The *csgA* gene encodes the CsgA pilin, the structural protein of curli fimbriae. The *csgB* gene encodes the CsgB minor subunit, and the *csgC* gene encodes the CsgC periplasmic protein, the function of which is unknown. However, it has been proposed that the CsgC protein plays a role in the secretion of the major protein of curli [60]. Briefly, through the NP, the CsgA protein is secreted into the outer membrane in soluble form, where the nucleator or minor subunit CsgB polymerizes monomeric proteins (CsgA) to form the amyloid structure [54].

#### 1.2.4. The Nonfimbrial Adhesin, TosA

TosA is a nonfimbrial adhesin with a molecular weight greater than 250 kDa that is closely related to UPEC. TosA is classified as a novel repeat-in-toxin (RTX) protein that is specifically expressed in the host UT and plays a crucial role in the virulence and survival of UPEC [61]. This adhesin has been associated with the phylogenetic group B2 and is a potential marker for strains with high content of virulence genes [62]. Interestingly, TosA is located in the outer membrane of *E. coli* but does not exhibit cytotoxic effects on mammalian cells. Instead, it enhances the adherence of *E. coli* to renal cell lines, contributing to pathogenesis by binding to receptors on the surfaces of host epithelial cells derived from the upper UT [63]. TosA facilitates three main functions: (i) adherence to host cells originating from the upper UT, (ii) survival during disseminated infections, and (iii) increased lethality during urosepsis [61,64]. Xicohtencatl et al., 2019, identified a relationship between strains carrying the gene that encodes the TosA protein, their virulence potential, resistance to both β-lactam and non-β-lactam antibiotics, and their enhanced adherence to the bladder cell line HTB-9 [62].

### 1.3. Immunity in UTIs

Three defense mechanisms against bacterial pathogens are present in the bladder: (1) physical barriers, (2) expressed peptides and proteins, and (3) immune cells [65]. The urothelium in the bladder is the first physical barrier and is made up of three to six layers of transitional epithelial cells. Umbrella cells constitute the first layer of cells in the lumen of the bladder and are coated with the protein uroplakin, which functions as a physical barrier to liquids, toxins, and microorganisms that do not express receptor proteins for uroplakin (UPK) and can adhere to the urothelium [66]. The umbrella cells are covered with mucus, composed of proteoglycans and glycosaminoglycans, and thus create a natural impermeable barrier [66]. The Tamm-Horsfall protein is a highly glycosylated protein, has many cysteines (disulfide bridges), is the most abundant glycoprotein in urine, and is produced exclusively in the tubular epithelial cells of the ascending limb of Henle [67]. The Tamm–Horsfall protein competes with uroplakin for interaction with FimH. This competition partially prevents the adherence of uropathogens that have fimbrial proteins, including FimH, which recognize and bind to mannosylated uroplakins. The structure of the Tamm–Horsfall protein is rich in mannose and contains disaccharides that bind to type I fimbriae, effectively competing with the mannose receptors on bladder epithelial cells (BECs). This action reduces the adhesion and colonization of UPEC in the bladder, facilitating its elimination through the urinary stream. Furthermore, the Tamm–Horsfall protein helps prevent excessive inflammation during bladder infection by inhibiting chemotaxis and reactive oxygen species (ROS) production by binding to the Ig-like lectin-9 (Siglec-9) receptor of neutrophils [68]. Secretory IgA (sIgA) is also produced in the UT; sIgA is produced by the plasma cells of the lamina propria and is subsequently secreted on the surface of the mucosa, in the intestine, and other organs [69]. The function of sIgA is to control the mucosal microbiota, limiting access to uropathogens and allergens; however, the function of sIgA in UTIs has not yet been fully elucidated [70]. The type and proportion of the immune cell population have been described; however, there are no reports describing the behavior of immune cells in the bladder. In contrast, other studies have investigated immune system behavior in murine models of bladder cancer [71]. Flow cytometry has been used to show that healthy murine bladders contain between 30,000 and 50,000 CD45^+^ cells, with 100-fold variation depending on organ dissociation [72,73]. Several studies have shown that the majority of immune cells in the bladder (70%) are antigen-presenting cells, where F4/80^+^CD64^+^ (40%) are the most abundant macrophages, followed by CD11b^+^ (15%) and CD130^+^ (5%) dendritic cells (DCs). The remaining 30% of immune cells are natural killer (NK) cells, mast cells (MCs), CD4 + ɑβ T cells, and γδ T cells [73].

#### Vaccines Targeting UTIs

Antibiotics are the primary treatment for UTIs, despite the emergence of MDR and XDR UPEC strains, which makes treatment difficult. The development of vaccines, potentially prophylactic vaccines, may be a good alternative to support the treatment of UTIs with antibiotics [74]. The Uromune vaccine was developed in Spain. Uromune is administered sublingually, with a required daily dose for two consecutive days. Each dose contains 100 μL of a culture of 108 inactivated bacteria (*E*. *coli*, *Klebsiella pneumoniae*, *Proteus vulgaris*, and *Enterococcus faecalis*). The efficacy of this vaccine for protection against UTIs is 63.3% to 81% over six months and 56.6% to 90.3% at 15 months [75].

The Urovaxom vaccine, also called OM-89, was developed in Switzerland and is presented in tablets containing 6 mg of lyophilized bacterial lysates from 18 different strains of *E*. *coli*. The indicated dose is one tablet daily for three months, with four subsequent booster doses. One tablet is administered daily during the first ten days of months 6 to 9. Several studies have shown an efficacy of 52.6% to 87.5% for protection against UTIs; however, side effects such as headaches, dizziness, nausea, erythema, and occasionally hair loss have been reported [76] (Table 1). Solco-Urovac is a vaccine that was developed in Germany. Solco-Urovac is administered as a vaginal suppository, and each suppository contains ten strains of different inactivated uropathogens (six different strains of *E. coli*, one strain of *K. pneumoniae*, one strain of *Proteus mirabilis*, one strain of *Morganella morganii* and one strain of *E. faecalis*). The indicated dose is one suppository per week for three weeks, with three booster doses involving the administration of one suppository per month. Studies have showed 22.2% to 25% protection against UTIs after six months of administration without boosters and 46% to 55.6% protection when boosters are administered. However, side effects, such as burning sensation, low grade fever, nauseas, vaginal rash, and vaginal bleeding, have been reported [77] (Table 1). ExPEC4V is a genetically modified vaccine developed by Johnson & Johnson that uses endotoxin A from *Pseudomonas aeruginosa* bound to four surface polysaccharides (serogroups O1A, O2, O6a, and O25b of *E. coli*). Intramuscular administration of ExPEC4V has 52% efficacy in preventing UTIs. The reported side effects of this vaccine are headache, dizziness, chills, diarrhea, limb pain, hyperhidrosis, and persistent pain at the injection site [78] (Table 1). Briefly, commercial vaccines for UTIs have demonstrated promising protection results; however, there are also reports of adverse outcomes with no noticeable improvement in patients [79]. A key advantage of these vaccines is their ability to reduce the number of UTI episodes, shorten the duration of antibiotic treatment, and help prevent the development of MDR strains that are tolerant to antibiotics. The use of vaccines in UTIs caused by MDR bacteria has been controversial, although more benefits have been obtained in reducing symptoms [78]. Importantly, their use for the general population is difficult owing to the high costs and the fact that they do not provide 100% protection. Unfortunately, the use of commercial vaccines in pediatric patients has not been accepted to date.

Importantly, current vaccines against UTIs have not provided significant protection and have caused various adverse effects. Recently, our working group developed a double and triple fusion of the main adhesins (FimH, PapG, and CsgA) of UPEC, with a focus on eradicating or reducing the incidence of UTIs. The dimeric protein FC [FimH(F) + CsgA(C)] and trimeric protein FCP [(FimH(F) + CsgA(C) + PapG(P)] can capacity to promote the release of 464.79 pg/mL and 521.24 pg/mL of IL-6, respectively [74]. Additionally, the FC protein triggered the release of 398.52 pg/mL of IL-8, and the FCP protein triggered the release of 450.40 pg/mL of IL-8. Inhibition data have shown that anti-FC and -FCP antibodies block the adhesion of CFT073 to HTB-5 cells by 73% and 46%, respectively [74]. Our results suggest that these fusion proteins can function as protective molecules against UTIs. Other studies have shown that the FimH and FliC proteins trigger high concentrations of IL-6 and IL-8 when coculture models of HTB-5 and HMC-1 cells are used; however, these cytokines are detected at low concentrations in the presence of the CsgA protein [80].

### 1.4. UPEC Genetic Diversity

Genetic diversity in UPEC strains has been determined mainly by pulsed-field gel electrophoresis (PFGE), multilocus sequence typing (MLST), and recently, by next-generation whole-genome sequencing (NGS) [81]. Sequence typing is an essential tool for the identification of clonal lineages and clonal complexes (CCs) in UPEC strains; the Achtman-Oxford MLST is the most widely used method [82,83]. Meta-analysis studies have revealed the presence of 20 sequence types (STs) (ST131, ST69, ST10, ST405, ST38, ST95, ST648, ST73, ST410, ST393, ST354, ST12, ST127, ST167, ST58, ST617, ST88, ST23, ST117, and ST1193) in extraintestinal *ExPEC* strains [60]. Our workgroup identified 14 STs associated with UPEC strains of serotypes O25b, ST73, ST93, ST120, ST405, ST998, ST10, ST62, ST443, ST117, ST421, ST69, ST10, ST38, and ST95, all of which were recovered from Mexican children with complicated UTIs [23].

#### 1.4.1. Main STs in UPEC

ST131 lineage. ST131 is an important lineage of UPEC that is recognized globally and is associated with a wide range of virulence genes, high antibiotic resistance, and ESBLs [82,84,85]. The ST131 clone exemplifies the complex evolution that microorganisms can undergo, evolving from a commensal state to a multidrug-resistant state. Whole-genome sequencing studies have provided valuable insights into point mutations, critical genes selected due to antibiotic resistance, the presence of specific plasmids, genomic islands, and compensatory mutations [86]. ST131 clones responsible for UTIs are high-risk clones for resistance to multiple antimicrobial determinants and have MDR and XDR phenotypes [87]. High-risk clones are distributed globally and associated with several antimicrobial resistance determinants, which aid their transmission and persistence in hosts [88].

In addition, ST131 clones contain the *bla*_CTX-M-15_ gene, which confers resistance to cephalosporins. ST131 clones has been associated mainly with clinical UPEC strains belonging to serogroups O25b and O16. A specific subclone of UPEC known as ST131/O16, which is part of the ST131 clonal lineage, has emerged as a variant. This subclone has been identified in the fecal samples of both healthy individuals and patients suffering from cystitis and pyelonephritis in China [71,89]. High virulence has been observed in the ST131-O16 clones, associated with subcutaneous sepsis in murine models. This virulence is linked to a high frequency of the *fimH* gene and the presence of specific mutations in the *gyrA* gene (S83L and D87N) and the *parC* gene (S80I), which confer fluoroquinolone resistance [89,90]. Additionally, the *fimH*41 allele has been correlated with increased virulence in vitro [91]. In Nigeria, the ST131/O16 clone has been identified in patients diagnosed with sepsis [92].

The UPEC/ST131 clones of serogroup O25b (UPEC/ST131/O25b) is the most prevalent clone identified in adult women in UTIs across China. This clone is resistant to fluoroquinolones due to specific mutations in the *gyrA* (S83L, D87N, and A93E/G) and *parC* (S80I and E84N) genes, as reported by Zhong et al. (2019) [90,93]. Additionally, mutations in the *gyrB* gene have also been associated with fluoroquinolone resistance [94]. Furthermore, the ST131-O25b clone has been observed to carry the *fimH*30, *fimH*41, and *fimH*27 alleles, according to findings by Dahbi et al. (2014) [95].

Single-nucleotide polymorphism (SNP) studies have revealed three alleles or variants of *fimH* (H30, H30-R, and H30-Rx) in clones of UPEC/ST131 [96,97]. Interestingly, clones of ST131 have diversified into three clades (A, B, and C) on the basis of variations in the fimbrial adhesin gene *fimH* [96,97]. Clade A corresponds to the *fimH*41 variant, which has been suggested to have originated in Southeast Asia. Clade B is characterized by the *fimH*22 variant, whereas clade C is characterized by the *fimH*30 variant, both of which originated in North America [96]. Additionally, clade C includes two subclades: subclade C1, which contains the H30-R subclone, and subclade C2, which includes the H30-Rx subclone [97,98]. Strains with fluoroquinolone resistance and carrying the *bla*_CTX-M-15_ gene were identified in subclone H30. Subclone H30-R is is loaded with strains that are resistant mainly to fluoroquinolones, and subclone H30-Rx contains mainly the *bla*_CTX-M-15_ gene [98,99]. The C1/H30-R subclade is defined by double mutations in the *gyrA* (S83L and D87N) and *parC* (S80I and E84V) genes [93], along with the presence of the *bla*_CTX-M-14_ or *bla*_CTX-M-27_ genes. Within the C1/H30-R subclade, there is a sublineage called C1-M27, which is specifically characterized by the *bla*_CTX-M-27_ gene and the F1:A2:B20 plasmid [100]. In contrast, the C2/H30-Rx subclade also presents mutations in the *gyrA* and *parC* genes, but it is associated with the *bla*_CTX-M-15_ gene and the F2:A1:B-plasmid. Additionally, the C2/H30-Rx subclade harbors several virulence genes (*iutA*, *afa*, *dra*, and *kpsII*), the K100 capsule, multidrug resistance genes, and genes responsible for ESBL production [96,101,102]. Serogroup O25b is associated mainly with ST131, is widely distributed throughout the world, and is a high producer of ESBLs in children and adults [23,103,104,105,106,107,108,109,110]. In addition, the genes encoding class C β-lactamases, DHA-4 and CYM-42, are involved in UTI processes in Tunisia [111] (Figure 1).

##### ST1193 Lineage

The emerging ST1193 lineage was first identified in 2007 and has been associated with systemic infections and UTIs, as well as with companion animals and the environment [114]. This clone has been isolated from fecal matter and is associated with bloodstream infections and UTIs [115]. In the United States, UPEC clones with ST1193 from the urine of adults (<40 years) were found to be resistant to fluoroquinolones, trimethoprim/sulfamethoxazole, and tetracyclines [116]. In Japan in 2020, ESBL-positive UPEC strains (*bla*_CTX-M-27_, *bla*_CTX-M-14_, and *bla*_CTX-M-15_) were associated with ST1193 [117]. ST1193 belongs to clonal complex 14 (CC14), which is related with the serogroup O75 clonal group; is a nonlactose fermenter; is related to phylogenetic group B2; and is resistant to quinolones [114,118,119], maintaining a correlation with ST131 clones [119,120,121]. Several virulence genes associated with ST1193 clones have been described, such as *papA* (pili P main subunit), *iha* (adhesin and siderophore), *fimH* (type 1 fimbriae adhesin), *sat* (secreted autotransporter toxin), *vat* (vacuolizing autotransporter toxin), *fyuA* (yersiniabactin), *kpsMII* (capsule synthesis), *usp* (uropathogenic specific protein), *ompT* (outer membrane protein), and *malX* K1/K5 (production of K1- and K5-type polysaccharide capsules) [115]. The UPEC/ST1193 clones from uncomplicated UTIs (unUTIs) of women in Spain is resistant to fluoroquinolones and belongs to the O75:H4 serotype [120]. In Mexico, UPEC/ST1193 clones have been observed in women with rUTIs of several years’ duration [121]. The genomes of these strains contained chromosomal mutations that conferred resistance to fluoroquinolones (*gyrA*, S83L, D87N, *parC* S80I, and *parE* L416F) [122]. The genes *chuA* (heme group receptor), *fyuA* (yersiniabctin), *gad* (glutamic acid dehydratase), *iha* (adhesin and siderophore), *irp2* (regulation of iron acquisition), *iucC* (aerobactin biosynthesis), *iutA* (aerobactin), *kpsE* (polysaccharide transport), *kpsMII* K1 (capsule synthesis serogroup K1), *neuC* (UDP-GlcNAc 2-epimerase), *ompT* (outer membrane protein), *papA_F43* (main subunit of pili P F43), *sat* (secreted autotransporter toxin), *senB* (shiga-like toxin), *sitA* (iron transport system), *terC* (tellurite resistance protein), *tratT* (complement resistance protein), *usp* (uropathogenic specific protein), *vat* (vacuolizing autotransporter toxin), and *fcV* (iron and cobalt transport protein) have been identified [112].

In Vietnam, ST1193 clones from sepsis patients and from fecal matter presented a high frequency of genes encoding adhesins *papG* (type P fimbriae adhesin), *fimH* (type 1 fimbriae adhesin), and *csg*A (curli fimbriae adhesin); toxins *sat* (secreted autotransporter toxin), *usp* (uropathogenic specific protein), and *senB* (shiga-like toxin); siderophores *fepA* (enterobactin receptor), *chuA* (heme group receptor), *fyuA* (yersiniabctin), *irp1* (yersiniabactin biosynthesis), *sitA* (iron and manganese transport), *iucA* (aerobactin biosynthesis), and *iutA* (aerobactin); and invasins *ibeB* (invasine), *vat* (vacuolizing autotransporter toxin), and *hlyE* (hemolysin) [123]. ST1193 clones also contain the IncF−:A1:B10 plasmid located within the C1 cluster, as well as the IncF1:A2:B20 plasmid in ST131 clones, both of which contain the conjugation module, indicating that the ST1193 clone is a self-transferable plasmid [123]. The similarity of STs in several countries (Vietnam, Australia, the United States, Denmark, and Singapore) indicates that there is a high transfer of genes and plasmids that contribute to the pathogenesis of these bacteria [88,115,124,125].

Genome sequence data have shown that Mexican strains of UPEC identified as ST1193 lack the genes *fimH* (type 1 fimbriae adhesin), *papG* (type P fimbriae adhesin), *hlyA* (hemolysin), and *satA* (secreted autotransporter toxin). However, these strains harbor four plasmids (small cryptic, nonconjugative mobilization, conjugative mobilization, and resistance plasmids) associated with virulence and resistance [126]. Interestingly, the ST1193 clone has high activity in promoting adherence to the human bladder cell line HTB-5 and nonbiofilm formation [126] (Figure 1). ST131 and ST1193 clones of UPEC pose a significant risk to human health because of their resistance to multiple antibiotics and widespread global distribution. Both clones developed this resistance through mutations in the QRDR region, which is responsible for quinolone resistance. Additionally, they can carry IncF plasmids and virulence factors, contributing to their prevalence among *E. coli* isolates [112,127].

##### ST73 Lineage

The prevalence of UPEC clinical strains identified with ST73 is significant, especially in the United Kingdom, Spain, Switzerland, and Portugal. The pathogenic potential of these clones has been linked to the production of ESBLs that confer multidrug resistance and can cause sequelae of urosepsis and bacteremia [128]. In Mexico, 7.81% of ST73 clones have been reported from Mexican children with cUTIs [23]. ST73 clones belong to phylogroup B2 and are associated with four main serotypes (O6:H1, O2:H1, O18:H1, and O25:H1) and several virulence genes: *fimH* (type 1 fimbriae adhesin), *papG* (type P fimbriae adhesin), *csgA* (curli fimbriae adhesin), *expA* (synthesis and secretion of exopolysaccharides), *fdeC* (fructose transport protein), *iut* (aerobactin transport protein), *fyuA* (yersiniabactin), *chuA* (heme group receptor), *hma* (siderophore), *sit* (sit iron transport system), *eit* (iron-induced enterotoxin), and *ire* (siderophore) [128,129]. Significant variations among strains of different serogroups, including ST73/O25 clone, have revealed many toxin genes, such as *cnf1* (cytotoxic necrotizing factor), *hlyABCDE* (hemolysin), and *sat* (secreted autotransporter toxin), and genes that promote bloodstream dissemination and invasion, *pic* (proinflammatory cytokine-induced cytotoxin), *tsh* (temperature-sensitive hemagglutinin), *iss* (serum resistance protein), *ibe* (blood–brain barrier invasion protein), *hek* (hemagglutinating adhesin), and *tia* (subtilase) [128]. Interestingly, the *tcp* gene associated with immune modulation via inhibition of Toll/IL-1 receptor signaling was identified in ST73/O6 clones [129]. Furthermore, ST73 clones contain class 1 integrons (*intl*1) and *bla*_OXA-1-type_ β-lactamases [129].

##### ST405 Lineage

The ST 405 clone associated with phylogroup D, are resistant to quinolones (*gyrA*, *S83L* and *D87N*; *parC*, *S80I*; and *parE*, *S458A*) [130] and to ESBLs (*bla*_CTX-M-14_ and *bla*_CTX-M-15_); however, some ST405 clones harbor the *bla*_CTX-M-27_ gene, as has been identified in other STs. ST405 clones shares virulence and multidrug resistance characteristics with clones belonging to ST131/O25b [131] (Figure 1).

##### ST69 Lineage

Clones belonging to ST69 are associated with phylogenetic group D; are resistant to trimethoprim/sulfamethoxazole; are considered endemic; and have a wide variety of serogroups, such as O11, O15, O17, O44, O73, O77, O86, O125ab, and O25b [5,132]. Our workgroup revealed that ST69 clones was highly prevalent in children with cUTIs, with high loads of virulence genes (*fimH*, *csgA*, *papGII*, *ecpA*, *iutD*, *fyuA*, and *hlyA*) and resistance genes (MDR and XDR), as well as 21.81% of the strains belonging to serogroup O25 [23] (Figure 1).

##### Other STs

ST8196 is a new ST identified in clones from patients with bacteremia that phylogenetically shares a common ancestor with ST131 [113]. Clones belonging to ST101 carry the *bla*_NDM-7_ gene and are associated with the resistance genes *bla*_OXA-1_, *bla*_TEM-1A_, *bla*_CTX-M-15_, *aac(6′)*-*Ib-cr*, *catB3*, and *tetB* [133]. As of 2019, 29 variants of NDM-type enzymes have been reported, which are distributed mainly in countries such as China and India [134,135,136]. Briefly, phylogenetic studies have shown that the adaptation of new STs in *E. coli*, with virulence and resistance determinants, represents a health problem, and epidemiological studies at the molecular level are needed. The ST410 clone has been identified mainly in ExPEC and is becoming increasingly significant as a high-risk clone. Research indicates a high frequency of transmission between species [88,137]. Whole-genome sequencing of 10 isolates from both animals and humans revealed a high degree of genetic similarity, distinguished by a low number of SNPs. The ST410 clone has been identified predominantly in ExPEC and is increasingly recognized as a high-risk clone. Research has demonstrated a significant frequency of interspecies transmission (humans, poultry, companion animals, and sewage) in Germany [137,138]. The ST410 clone has been reported in several countries, including China, Italy, Denmark, and Ghana [136,139,140,141]. Clones of ST410 carry the *bla*_OXA-181_ gene, an OXA-48-type carbapenemase that confers resistance to penicillins and carbapenems. The *bla*_OXA-181_ gene is believed to have originated from *Shewanella xiamenensis*, which is an environmental bacterium [142]. In addition, there are reports of its identification from a natural water source in Singapore with resistance to carbapenems, as well as its similarity with isolates in Thailand. Furthermore, the IS26 insertion sequence has been mentioned as a mediator of the acquisition of resistance through the *bla*_NDM-5_ gene [143] (Figure 1).

## 2. Conclusions

UTIs continue to have a high impact worldwide due to the increased difficulty of antimicrobial treatment caused by the emergence of MDR and XDR strains. The acquisition of antibiotic resistance genes and the dissemination of virulence factors have facilitated recurrence (Figure 2). A variety of clinical treatment alternatives to antibiotic therapy have been developed, such as the use of vaccines, although the protective efficacy of these vaccines has not been achieved. Moreover, the high diversity of serogroups has limited the vaccine protection spectrum, and these vaccines have been associated with multiple side effects, prolonged use times, and high costs. Therefore, it is imperative to study the genetic diversity of UPEC and the immune response that develops in the host during a UTI to develop new and better treatment options that reduce the impact of UTI recurrence and persistence.

## 3. Perspectives

The pathogenic cycle of UPEC in the urinary tract begins with the expression of various colonization factors, which are essential as a first step for pathogenicity. The adhesins CsgA, FimH, and PapG are commonly found across different UPEC lineages and play crucial roles in the initial colonization process of the urinary tract. This step is fundamental to the pathogenic cycle of bacteria. Therefore, a cutting-edge therapeutic strategy should focus on creating a chimeric fusion protein that combines these three adhesins. This fusion protein could serve as a specific vaccine against UPEC UTIs by generating protective memory antibodies that reduce or block initial adherence to the bladder epithelium (Figure 2).

## Figures and Tables

**Figure 1 microorganisms-13-00745-f001:**
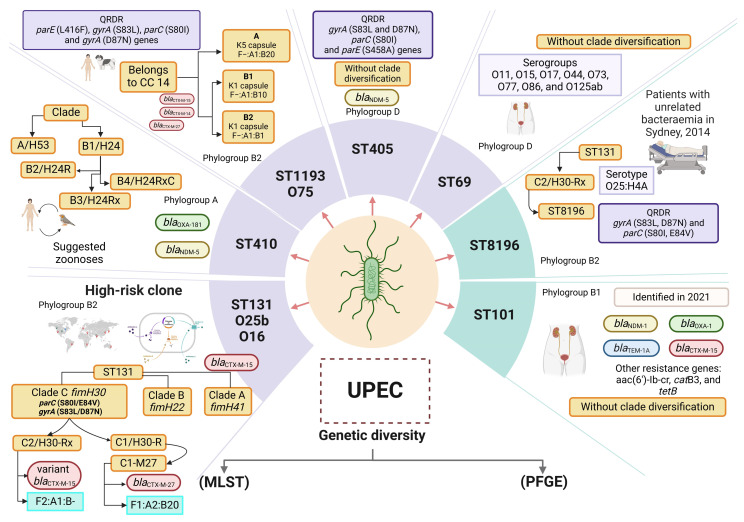
Genetic diversity of UPEC. Sequence type (ST) analysis using multilocus sequence typing (MLST) and pulsotype analysis with pulsed-field gel electrophoresis (PFGE) are two widely used methods for the analysis of genetic diversity in UPEC. PFGE pulsotype analysis has allowed the evaluation of the bacterial genome of UPEC with macrorestriction assays using specific cuts with restriction enzymes. However, pulsotypes are highly diverse, and pulsotyping cannot be used to determine the characteristics of the various clonal lineages of UPEC. In addition, ST analysis by MLST has been used to determine the associations of virulence and resistance characteristics with the most important clonal lineages of UPEC. The ST8196 clone has been reported as a sublineage within ST131, which shares the same ancestor. The most important ST lineages in UPEC are shown in light purple squares, and the recently reported STs are shown in light green squares. The *bla*_CTX_ genes are indicated in light red ovals, the *bla*_OXA_ genes are indicated in light green ovals, the *bla*_NDM_ genes are indicated in light yellow ovals, and the *bla*_TEM_ genes are indicated in light blue ovals. Each ST shows the representative characteristics of each lineage/ST, serogroup, subclone, and phylogenetic group and the presence of genes associated with the antibiotic hydrolysis resistance mechanism. Two clonal lineages have been identified among the clones from ST410: clade A/*fimH*53 and clade B/*fimH*24. The B/H24 lineage has been further divided into three subclades: B2/H24R (which exhibit fluoroquinolone resistance); B3/H24Rx (which carries the *bla*_CTX-M-15_ gene); and B4/H24RxC (which contains the *bla*_OXA-181_ gene) [83]. The ST1193 clone has diversified into two clades: ST1193-A (characterized by the K5 capsule and F−:A1:B20 plasmids) and ST1193-B (features the K1 capsule). The ST1193-B clade is further divided into subclades ST1193-B1 (with the plasmid F−:A1:B10) and ST1193-B2 (with the plasmid F−:A1:B1 [112]. The ST8196 clone is associated with phylogroup B2 and has the serotype O25:H4A. It is derived from the ST131/C2-H30Rx clone [113]. Design of this study. Created with http://www.Biorender.com.

**Figure 2 microorganisms-13-00745-f002:**
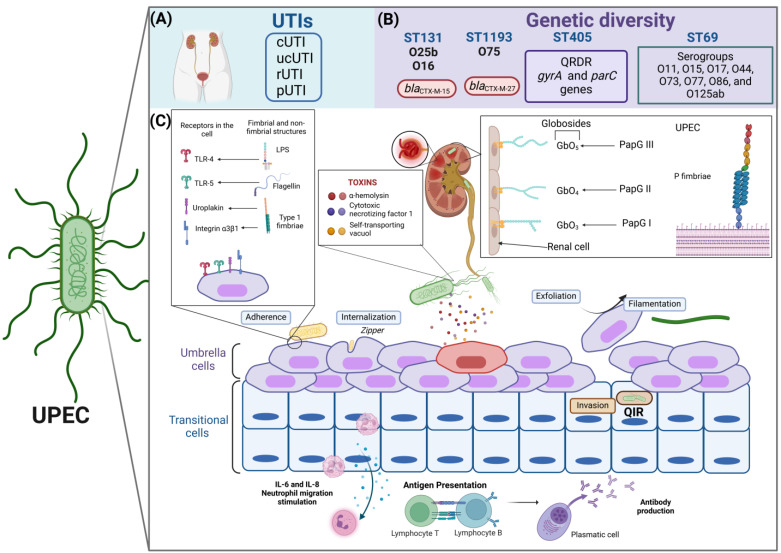
Characteristics of the pathogenic cycle of UPEC, type of infection, and genetic diversity. (**A**) UPEC is the main uropathogen associated with urinary tract infections (UTIs) and can be recovered from different clinical stages, such as the complicated (cUTI), uncomplicated (unUTI), recurrent (rUTI), and persistent (pUTI) stages. (**B**) UPEC shows high clonality, highlighting that the high-risk clones, such as ST131/O25b, ST131/O16, and ST1193/O75, and other clones that have been frequently reported in isolates of urinary origin are ST69 and ST405. UPEC clones present particular characteristics of antibiotic resistance, which allows their identification as epidemiological risk clones. (**C**) The pathogenesis of UPEC in the bladder epithelium begins with the specific adherence of UPEC type 1 fimbriae to the mannosylated residues of the uroplakin receptors and integrin α_3_β_1_-type receptors on the surface of the umbrella cells of the urothelium. Curli fimbriae, which are recognized by fibronectin and laminin in the cell, also participate. Through a rearrangement of the actin filaments, UPEC is internalized in superficial cells or umbrella cells via a zipper-type mechanism. Once inside the cell, UPEC can form an intracellular bacterial community (IBC) or carry out a filamentation process in which UPEC maintains bacterial growth but does not divide so that it can form filaments, which facilitate greater contact with the umbrella cells and allow a new cycle of infection to begin. Exfoliation of the urothelium occurs naturally or is favored by external signals caused by UPEC, such as the release of α-hemolysin (HlyA), cytotoxic necrotizing factor 1 (Cnf-1), and vacuolating self-transporting toxin (Sat), which cause cell damage, inducing apoptosis and excessive exfoliation. After the exfoliation process, UPEC can infect transitional epithelial cells, forming quiescent intracellular reservoirs (QIRs). The nonfimbrial structures of UPEC, such as lipopolysaccharides and surface glucans, are recognized by TLR-4 receptors on the surface of the bladder, and TLR-5 recognizes flagellin. P fimbriae are essential virulence factors for colonizing the renal epithelium and are also recognized by the major histocompatibility complex type 1 of antigen-presenting cells. The P fimbriae variants PapGI, PapGII, and PapGIII have affinities for globosides in renal cells. Finally, UPEC can evade the host immune response and the action of antimicrobials during its pathogenic cycle, which is thus associated with recurrent and persistent infections. UPEC synthesizes the flagellum to colonize the upper UI in an ascending manner, reaching the kidney, where it can cause pyelonephritis and, subsequently, reach the hematogenous route to cause urosepsis or septicemia, created with http://www.Biorender.com (design in this study).

**Table 1 microorganisms-13-00745-t001:** Commercial vaccines for the treatment of UTIs.

Vaccines	Composition	Administration and Doses	Potential Adverse Effects (Not Severe or Fatal)	References
Urovaxom (OM-89)	6 mg bacterial lyophilisate of 18 strains of *E. coli.*	The dose is one capsule per day for three months, followed by reinforcement of one capsule every ten days for six to nine months.	Headache, dizziness, nausea, erythema, and decreased hair growth. ≥5% (n = 451)	[76]
Uromune	Inactivated bacterial concentrate of *E. coli*, *K. pneumoniae*, *P. vulgaris*, and *E. faecalis.*	Two shots (10^8^ bacteria/shot) once a day for three months.	Nausea, erythema, intermittent abdominal pain, pruritus over the abdomen, and postnasal drip. <10% (n = 77).	[75]
Solco-Urovac	Inactivated bacterial concentrate of six strains of *E. coli*, *K. pneumoniae*, *P. vulgaris*, *M. morganii,* and *E. faecalis.*	One vaginal suppository once a week for three weeks and then a reinforcement of one suppository per month for three months.	Burning sensation, low grade fever, nauseas, vaginal rash, and vaginal bleeding <10% (n = 75).	[77]
ExPEC4V	Modified exotoxin from *P. aeruginosa* bound to four *E. coli* surface polysaccharides (serogroup O1A, O2, O6a, and O25b).	Single dose intramuscular injection of 0.5 mL.	Headache, dizziness, chills, fever, dysgeusia, pain in the upper abdomen, diarrhea, hyperhidrosis, and pain in the extremities. <60% (n = 93).	[78]

## Data Availability

No new data were created or analyzed in this study.

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
