# Peer review of "Pathogenesis and Immunomodulation of Urinary Tract Infections Caused by Uropathogenic Escherichia coli"

_microorganisms, 2025, doi:10.3390/microorganisms13040745_

Round 1
Reviewer 1 Report
Comments and Suggestions for Authors
In this review, the authors describe major adhesins and the diversity of Escherichia coli clones and their specific properties that can cause urinary tract infection. They further discuss the immune defense and composition of immune cells associated with the urinary tract and vaccine strategies, their efficacy and their side effects. They also partly focus on the description of these features in children.
Major comments
The abstract does not reflect the content of the review. For example, vaccine development and the certain focus on UTI in children is not mentioned.
There is no discrimination between lower and upper UTI and the distinct strain populations that cause these infections.
The descriptions of adhesins and equally of major clones causing urinary tract infection is not well described and organized. For example, it is not clear that FimH is a fimbrial adhesin on the tip of type 1 fimbriae and to which structures FimH is binding. For the bacterial clones, nosystematic information is given in Figure 1. To the knowledge of this reviewer distinct amino acid substitutions occur in GyrA, GyrB and ParC irrespectively of clonal type upon the acquisition of fluoroquinolone resistance.
The findings of the authors are not well embedded in the review.
A focus on recent, novel findings not yet covered in other reviews would be beneficial.
Other comments (not comprehensive)
l.26: TosA is only mentioned in the abstract.
Introduction: The focus is on pediatric UTI, why? And what are the specific differences between UTI in children vs. adults?
Nucleation-precipitation assembly; Why is only csgBAC discussed?
Table 1: what is the frequency and severity of the site effects? Is it similar to all vaccines?
Major STs in UPEC: The description does not reflect the well investigated development of the ST131 lineage from a commensal to a multidrug resistant pathogen. Here and in other descriptions of clones different terms are indiscriminatory used without definition and providing information about the level of resolution. Further serogroups can be variable and do not necessarily reflect the phylogeny of the isolates. H30 is not a clone, but a fimH allele and without mentioning its association with the subclade this information is not in a context. Also ST131 can harbor the blaCTX-M-15 or the e blaCTX-M-14 gene dependent on the clade. blaCTX-M-15 does not mediate fluorequinolone resistance. ,,,
L. 263: What are these virulence genes doing? Are all adhesins?
l.277:Most or all of these clonal lineages are pandemic.
Author Response
REVIEWER 1
- The abstract does not reflect the content of the review. For example, vaccine development and the certain focus on UTI in children is not mentioned.
R/ We appreciate Reviewer 1's comments, which have greatly enhanced this section of the manuscript. The changes made to the abstract are detailed below:
Line 33 to 36. New knowledge regarding protein structures known as adhesins and their role in the infection process can help identify therapeutic targets and aid in the design of vaccines. These vaccines could be based on developing chimeric fusion proteins (FimH+CsgA+PapG), which may significantly reduce the incidence of UTIs in pediatric and adult patients.
- There is no discrimination between lower and upper UTI and the distinct strain populations that cause these infections.
R/ We appreciate Reviewer 1's comments, which we have addressed. The following information has been added to the manuscript:
Lines 84 to 90. UTIs can be classified based on their anatomical site into two categories: lower urinary tract infections (lUTI) and upper urinary tract infections (uUTI). lUTI includes conditions such as cystitis, while uUTI encompasses urethritis, pyelonephritis, and urosepsis. The uropathogens responsible for causing lUTI include Escherichia coli, Staphylococcus saprophyticus, Staphylococcus spp., Proteus spp., Klebsiella spp., Enterococcus spp., Pseudomonas spp., and various yeast species. Notably, E. coli accounts for over 80% of the uropathogens associated with uUTI. More than 80% of community-acquired UTIs and 20–40% of hospital-acquired UTIs are associated with uropathogenic Escherichia coli (UPEC).
- The descriptions of adhesins and equally of major clones causing urinary tract infection is not well described and organized. For example, it is not clear that FimH is a fimbrial adhesin on the tip of type 1 fimbriae and to which structures FimH is binding.
R/ We sincerely appreciate all the comments provided on this manuscript. Each suggested point has been thoroughly addressed to enhance the quality of the manuscript. The changes made are detailed below:
Lines 133 to 138.
1.2. Fimbrial and non-fimbrial adhesins in UPEC
Fimbria types 1 and P participate in the uropathogenesis of E. coli via Chaperone-Usher (CU) assembly mechanism (14, 15). In the CU system, chaperones are the proteins responsible for preventing the polymerization of structural proteins in the periplasm, keeping them soluble until their recognition by the accommodator protein, which catalyzes the correct folding of the pilins (15).
Lines 140 to 166.
1.2.1. Type 1 fimbriae. The expression of type 1 fimbriae has been linked to biofilm formation, adherence to and invasion of the uroepithelium, yeast agglutination, and immune response evasion by invading macrophages (16). Type 1 fimbriae contribute to the pathogenicity of E. coli, which is involved in the colonization of various niches from the intestine, genitals, and urinary tract to other organs (17). Type 1 fimbriae are encoded by the fimAICDFGH operon, which includes the regulatory genes fimB, fimE, and fimX and the fimS switch region. The FimA protein is a pilin that forms the fimbrial main subunit (filament), FimC is a periplasmic chaperone, and FimD is an accommodator protein. The FimI protein is a homolog of FimA; the function of FimI is unknown, but FimI is essential for the expression of fimbrial proteins (18). The tip of the fimbria comprises the docking proteins FimF and FimG, which are involved in binding a single copy of the FimH adhesin. The FimH adhesin contains two domains: the pilin domain, which interacts with the FimG protein, forming a flexible junction, and the lectin domain, which forms a binding pocket for a receptor that recognizes mannosylated residues (19). The expression of type 1 fimbrial proteins is regulated by two site-specific recombinases, FimB and FimE, which promote the inversion of the fimS switch region in response to stress factors that positively or negatively stimulate the expression of this fimbrial protein (20). Our working group described the frequency of type 1 fimbriae in UPEC strains from complicated UTIs [92.1% (164/178)] and of clone 025b/ST131 [95.2% (120/126)] in pediatric patients (13-14).
Lines 168 to 239.
1.2.2. Fimbria P. The distal part of pyelonephritis-associated fimbriae is anchored to the PapG adhesin, which interacts with glycosphingolipids consisting of Gal-α-(1,4) Gal residues. Three alleles have been described: PapGI, PapGII, and PapGII, which bind, respectively, and differentially, to the following globoside variants: GbO3, GbO4 (both abundant in human uroepithelial cells), and GbO5 (abundant in canine cells but not in human cells) (11). PapG variations in clinical strains of UPEC are an adaptation mechanism of bacteria that allows high adaptability to colonize and cause infection in humans efficiently (21).
The P fimbriae is encoded in the papBAHCDJKEFG operon and is clustered in the pathogenicity island (PAI) called PAI-CFT073-pheV (PAI ICFT073). A transcriptional repressor, called PapI, regulates the transcription of the P fimbria, and a transcriptional activator, called PapB. The PapA protein is the pilin that forms the major subunit, PapH is the protein that forms the minor subunit, PapC is the accommodator protein, PapD is the chaperone, and PapG is the adhesin (22). Recently, the prevalence of pap alleles was reported to be 34.3% (61/178) for papGII and 1.7% (3/178) for papGIII in a collection of UEc strains. Interestingly, a higher prevalence was observed from an EcO25b collection; 80.9% (102/126) amplified the papGII gene, and 2.4% (3/126) amplified the papGIII gene(13-14).
Lines 241 to 259.
1.2.3. Fimbria curli. Amyloid structures such as curli fimbriae use an extracellular assembly mechanism involving the NP, also called the type VIII secretion system. A wide variety of microorganisms produce functional amyloid components as structural support that favors the integrity of biofilms, a mechanism that promotes the colonization of abiotic and biotic surfaces (23). Curli fimbriae are made up mainly of the CsgA protein, a fine structure with aggregation and adherence properties (24). The genes encoding the proteins involved in curli biogenesis are located in the csgBAC operon, which is transcribed divergently. The csgA gene encodes the CsgA pilin, the structural protein of curli fimbriae. The csgB gene encodes the CsgB minor subunit, and the csgC gene encodes the CsgC periplasmic protein, the function of which is unknown. However, it has been proposed that the CsgC protein plays a role in the secretion of the major protein of curli (25). Briefly, through the NP, the CsgA protein is secreted into the outer membrane in soluble form, where the nucleator or minor subunit CsgB polymerizes monomeric proteins (CsgA) to form the amyloid structure (24).
Lines 261 to 287.
1.2.4. Non-fimbrial adhesin called TosA. TosA is a non-fimbrial adhesin with a molecular weight greater than 250 kDa, which is closely related to UPEC. TosA is classified as a novel Repeat-In-Toxin (RTX) protein that is specifically expressed in the host urinary tract, playing a crucial role in the virulence and survival of UPEC (39). This adhesin has been associated with the phylogenetic group B2 and is a potential marker for strains with a high virulence potential. Interestingly, TosA is located in the outer membrane of E. coli but does not exhibit cytotoxic effects on mammalian cells. Instead, it enhances the adherence of E. coli to renal cell lines, contributing to pathogenesis by binding to receptors on the surfaces of host epithelial cells derived from the upper urinary tract (40). TosA facilitates three main functions: (i) adherence to host cells originating from the upper urinary tract, (ii) survival during disseminated infections, and (iii) increased lethality during urosepsis (39-42) identified a relationship between strains carrying the gene that encodes the TosA protein, their virulence potential, resistance to both β-lactam and non-β-lactam antibiotics, and their enhanced adherence to the bladder cell line HTB-9 (42).
- For the bacterial clones, nosystematic information is given in Figure 1. To the knowledge of this reviewer distinct amino acid substitutions occur in GyrA, GyrB and ParC irrespectively of clonal type upon the acquisition of fluoroquinolone resistance.
R/ Reviewer 1's comment is important and insightful. The STs section includes information regarding the reported mutations in the gyrA, gyrB, and parC genes, which are responsible for conferring resistance to fluoroquinolones in plates ST131, ST1193, and ST405. Additionally, Figure 1 has been updated with further information, as suggested.
Lines 429 to 452.
ST131. …in addition, ST131 clones contain the blaCTX-M-15 gene, which confers resistance to cephalosporins. ST131 has been associated mainly with clinical UPEC strains with serogroups O25b and O16. High virulence in O16 clinical strains has been associated with subcutaneous sepsis in murine animals, a high frequency of fimH, and the presence of specific mutations of the gyrA (S83L, D87N, and A93E/G), gyrB (S464Y and L422P), and parC (S80I and E84N) genes encoding fluoroquinolone resistance (64, 65). Serogroup O16 has been identified in the fecal matter of healthy patients, patients with cystitis and pyelonephritis (45).
Lines 494 to 691. ST1193…..The genomes of these strains contain chromosomal mutations that confer resistance of the bacteria to fluoroquinolones (gyrA, S83L; gyrA D87N; parC S80I and parE L416F) (Pitout et al. 2022). The genes chuA, fyuA, gad, iha, irp2, iucC, iutA, kpsE, kpsMII K1, neuC, ompT, papA_F43, sat, senB, sitA, terC, tratT, usp, vat, and fcV have been identified (66)….
Lines 867 to 871. ST405…. Strains belonging to ST405, associated with phylogroup D, are resistant to quinolones (gyrA, S83L/D87N; parC, S80I; and pare, S458A) (98) and to ESBLs (blaCTX-M-14 and blaCTX-M-15); however, some ST405 strains harbor the blaCTX-M-27 gene, as has been identified in other STs. ST405 strains share virulence and multidrug resistance characteristics with strains belonging to ST131-O25b (73) (Figure 1).
- The findings of the authors are not well embedded in the review.
R/ We appreciate the comments provided by Reviewer 1, which have greatly enhanced the manuscript. The modifications are detailed below:
In section 1.2. "UPEC Fimbrial Adhesins," we have added relevant information regarding the non-fimbrial adhesin called TosA, as studied by our research group. The new information is as follows:
Line 284 to 287. Xicohtencatl et al. (2019) identified a relationship between strains that carry the gene encoding for the TosA protein, their virulence potential and resistance to B-lactam and non-B-lactam antibiotics, and increased adherence to the bladder cell line HTB-9 (49).
Additionally, we have reorganized the studies conducted by our research group and placed them at the end of each corresponding section. This allows for a more comprehensive understanding, incorporating studies of strains from Mexican origin.
- A focus on recent, novel findings not yet covered in other reviews would be beneficial.
R/ The reviewer's feedback is valuable, is important comment that we focus on the pathogenesis and immunomodulation caused by UPEC strains during their infectious process in the urinary tract.
We highlight the roles of the three primary UPEC adhesins (FimH, PapG, and CsgA) as well as the non-fimbrial adhesin TosA, in the processes of colonization and invasion, biofilm formation, intracellular bacterial communities (IBCs), and quiescent intracellular reservoirs (QIRs). These adhesins may also provide essential insights for developing a viable vaccine against UTIs by UPEC.
Additionally, through our studies of sequence types (STs) and clonal complexes (CCs), we emphasize the significance of various clinical entities, including uncomplicated urinary tract infections (uUTIs), lower urinary tract infections (lUTIs), and recurrent urinary tract infections (rUTIs) in both children and adults. We have also addressed the genetic diversity of UPEC and made important changes, including more recent information that contributed to the improvement of the manuscript.
Other comments (not comprehensive)
- l.26: TosA is only mentioned in the abstract.
R/ We appreciate the feedback from Reviewer 1. Below is a description of the main features of the TosA adhesin:
Lines 261 to 287. 1.2.4. Non-fimbrial adhesin called TosA. TosA is a non-fimbrial adhesin with a molecular weight greater than 250 kDa, which is closely related to UPEC. TosA is classified as a novel Repeat-In-Toxin (RTX) protein that is specifically expressed in the host urinary tract, playing a crucial role in the virulence and survival of UPEC (39). This adhesin has been associated with the phylogenetic group B2 and is a potential marker for strains with a high virulence potential. Interestingly, TosA is located in the outer membrane of E. coli but does not exhibit cytotoxic effects on mammalian cells. Instead, it enhances the adherence of E. coli to renal cell lines, contributing to pathogenesis by binding to receptors on the surfaces of host epithelial cells derived from the upper urinary tract (40). TosA facilitates three main functions: (i) adherence to host cells originating from the upper urinary tract, (ii) survival during disseminated infections, and (iii) increased lethality during urosepsis (39-42) identified a relationship between strains carrying the gene that encodes the TosA protein, their virulence potential, resistance to both β-lactam and non-β-lactam antibiotics, and their enhanced adherence to the bladder cell line HTB-9 (42).
- Introduction: The focus is on pediatric UTI, why? And what are the specific differences between UTI in children vs. adults?
R/ Reviewer 1's comments are highly accurate. It is important to note that UTIs in pediatric patients are more challenging to diagnose. The review not only addresses pediatric UTIs but also includes information about UTIs in adults, highlighting the differences between the two. We have incorporated this information into the introduction, which now reads as follows:
Lines 65 to 73. Prompt identification of UTIs in children is essential for effective treatment, particularly because cystitis and pyelonephritis can often be hard to distinguish in children under two years old. In the United Kingdom, 80% of UTIs in children, mainly infants, can go unnoticed owing to the presence of nonspecific signs such as fever of 38°C or higher, irritability, vomiting, decreased appetite, and lethargy (4), and improper collection of urine samples (3). In recent years, recommendations have been made for the diagnosis, treatment, and follow-up of children with UTIs, as well as for imaging studies (renal and bladder ultrasound) to identify underlying urological anomalies and their consequences (3-5).
Lines 73 to 83. In adult patients, UTIs pose a significant health issue, mainly due to therapeutic failures linked to infections caused by MDR and XDR UPEC strains. In women, several risk factors contribute to the development of UTIs, including loss of estrogen, changes in vaginal microbiota, and increased vaginal pH. The signs and symptoms of a UTI in adults are typically clear and are characterized by cystitis, which includes dysuria (painful urination), polyuria (increased urination), urgent urination, and suprapubic pain. In cases of pyelonephritis, additional symptoms such as fever exceeding 38°C, lower back pain, and systemic symptoms may be present (7-9). Laboratory analyses with positive urine cultures (considering the patient's age), noninvasive urine sampling, and clinical history and evaluation are essential for an optimal diagnosis (6).
- Nucleation-precipitation assembly; Why is only csgBAC discussed?
R/ We appreciate the comments from Reviewer 1.
It is important to note that the biogenesis of amyloid-type fimbriae carried out by extracellular assembly via a process called nucleation-precipitation (NP). In UPEC, the curli fimbriae are of the amyloid type, with the main subunit being the CsgA protein. This CsgA protein is unique to UPEC and is the only protein assembled by this mechanism; therefore, it has only been discussed in the context of the csgBAC operon.
- Table 1: what is the frequency and severity of the site effects? Is it similar to all vaccines?
R/ We would like to thank Reviewer 1 for their comments on the manuscript.
In Table 1, we reviewed the percentages of adverse effects associated with each vaccine and added the necessary information. Based on the reports, we found that no severe or fatal adverse effects were observed with the use of these vaccines. The modified table is shown below:
|
Table 1. Commercial vaccines for the treatment of UTIs |
|
|
|||
|
|
Vaccines |
Composition |
Administration and doses |
Potential adverse effects (Not severe or fatal)
|
References |
|
|
Urovaxom (OM-89) |
6 mg bacterial lyophilisate of 18 strains of E. coli. |
The dose is one capsule per day for three months, followed by reinforcement of one capsule every ten days for six to nine months. |
Headache, dizziness, nausea, erythema, and decreased hair growth. ≥5% (n=451). |
[37] |
|
|
Uromune |
Inactivated bacterial concentrate of E. coli, K. pneumoniae, Proteus vulgaris, E. vulgaris.
|
Two shots (108 bacteria/shot) once a day for three months. |
Nausea, erythema, intermittent abdominal pain, pruritus over the abdomen, and postnasal drip. <10% (n=77). |
[36] |
|
|
Solco-Urovac |
Inactivated bacterial concentrate of six strains of E. coli, K. pneumoniae, Proteus vulgaris, Morganella morganii and E. faecalis.
|
One vaginal suppository once a week for three weeks and then a reinforcement of one suppository per month for three months.
|
Burning sensation, low grade fever, nauseas, vaginal rash, vaginal bleeding <10% (n=75). |
[38] |
|
|
ExPEC4V |
Modified exotoxin from Pseudomonas aeuruginosa bound to four E. coli surface polysaccharides (serotypes O1A, 02, O6A, and 025B). |
Single dose intramuscular injection of 0.5 mL. |
Headache, dizziness, chills, fever, dysgeusia, pain in the upper abdomen, diarrhea, hyperhidrosis, and pain in the extremities. <60% (n=93). |
[39] |
|
|
|
|
|||
- Major STs in UPEC:
At this point we have several comments.
- a) The description does not reflect the well investigated development of the ST131 lineage from a commensal to a multidrug resistant pathogen.
R/ We appreciate the reviewer's comments, which have improved the manuscript. We have revised and reorganized section 1.4.1. Main STs in UPEC and subsequent sections are used to identify and correct errors. Additionally, we have included brief information on the importance of clone ST131 as part of the commensal microbiota. The information is as follows:
Lines 421 to 427. ST131 lineage. ST131 is a important lineage of UPEC that is recognized globally and is associated with a wide range of virulence genes, high antibiotic resistance, and ESBLs (42-44). The ST131 clone exemplifies the complex evolution that microorganisms can undergo, evolving from a commensal state to a multidrug-resistant pathogen. Whole genome sequencing studies have provided valuable insights into point mutations, critical genes selected due to antibiotic resistance, the presence of specific plasmids, genomic islands, and compensatory mutations (63).
- Here and in other descriptions of clones different terms are indiscriminatory used without definition and providing information about the level of resolution.
R/ We agree with the comment. In all sections related to diversity, we reviewed and established a general criterion for cloning as the aggregate of genetically identical cells or organisms produced by a single progenitor cell.
- Further serogroups can be variable and do not necessarily reflect the phylogeny of the isolates.
We completely agree with the comment of Reviewer 1. Serogroups can vary among clinical strains, regardless of their ST. However, certain STs, particularly those associated with pandemics, have exhibited a predisposition towards specific serogroups. This characteristic can facilitate the study of clinical strains of E. coli.
- H30 is not a clone, but a fimH allele and without mentioning its association with the subclade this information is not in a context.
R/ The reviewer has made a very accurate observation. The paragraph referring to clone ST131 has been revised to highlight its diversity, considering the variation in the fimbrial adhesin gene fimH found in three clades: A, B, and C. We provide an explanation of each clade and their respective fimH variants. Additionally, we detail the subdivision of clade C into two subclades, C1/H30R and C2/H20Rx, focusing on their main characteristics. The modifications made to the ST131 are indicated below.
Lines 453 to 474. Single nucleotide polymorphism (SNP) studies have shown three alleles o variants of fimH (H30, H30-R, and H30Rx) in clinical strains of UPEC ST131.
Interestingly, clones of ST131 have diversified into three clades (A, B, and C) based on variations in the fimbrial adhesin gene, fimH. Clade A corresponds to the fimH41 variant, which has been suggested to have originated in Southeast Asia. Clade B is characterized by the fimH22 variant, while clade C is defined by the fimH30 variant, both of which originated in North America (65). Additionally, clade C includes two subclades: subclade C1, which contains the H30R subclone, and subclade C2, which includes the H30Rx subclone.
The C1/H30R subclade is defined by double mutations in the gyrA (S83L and D87N) and parC (S80I and E84V) genes (64), along with the presence of the blaCTX-M-14 or blaCTX-M-27 genes. Within the C1/H30R subclade, there is a sublineage called C1-M27, which is specifically characterized by the blaCTX-M-27 gene and the F1:A2:B20 plasmid (Tóth et al., 2024). In contrast, the C2/H30Rx subclade also exhibits mutations in the gyrA and parC genes, but it is associated with the blaCTX-M-15 gene and the F2:A1:B- plasmid. Additionally, the C2/H30Rx subclade harbors several virulence genes (iutA, afa, dra, and kpsII), the K100 capsule, multidrug resistance genes, and genes responsible for ESBL production (48, 49, 50).
- e) Also ST131 can harbor the blaCTX-M-15 or the e blaCTX-M-14 gene dependent on the clade.
R/ Reviewer 1's comment is very insightful. It is important to note that the literature indicates that the presence of the blaCTX-M-14 or blaCTX-M-27 genes characterizes ST131 strains belonging to the C1/H30R subclade. In contrast, ST131 strains from the C2/H30Rx subclade are associated with the presence of the blaCTX-M-15 gene (Matsuo et al. 2020, https://doi.org/10.1128/AAC.00202-20; Merino et al. 2018, https://doi.org/10.1093/jac/ dky296).
- f) blaCTX-M-15 does not mediate fluorequinolone resistance.
R/We appreciate the comment from Reviewer 1. This error has been corrected in the manuscript and now reads as follows:
Line 237. The following information “clones contain the blaCTX-M-15 gene, which confers resistance to fluoroquinolones” was replaced by “clones contain the blaCTX-M-15 gene, which confers resistance to cephalosporins”.
Line 245. The following information “clone H30” was replaced by “subclone H30”.
Line 246. The following information “clone H30-Rx” was replaced by “subclone H30-Rx”.
Line 247. The following information “H30-Rx contains” was replaced by “subclone H30-Rx contains”.
- L. 263: What are these virulence genes doing? Are all adhesins?
R/ Reviewer 1's observation is very accurate. As suggested by the reviewer, modifications have been made detailing each point.
- l.277: Most or all of these clonal lineages are pandemic.
R/ We appreciate the accurate information provided by Reviewer 1 and hope the following details correspond to the request. As the reviewer highlighted, the ST lineages included in this manuscript are pandemic and have been reported in various cities within the same region and countries across different continents. Additionally, the ST1193 lineage is considered emerging and is gaining greater importance worldwide. The manuscript has been read carefully, and modifications have been made to improve the management of the main ST lineages.
Line 477 to 479. The ST1193 lineage is considered to be emerging, was identified in 2007, has been associated with systemic and urinary tract infections, as well as in companion animals and the environment (82).
R/ Details regarding the ST410 have been included in the caption of Figure 1. It is read now as follows:
Line 899 to 901. Briefly, phylogenetic studies have shown that the adaptation of new STs in E. coli, with virulence and resistance determinants, represents a health problem, and epidemiological studies at the molecular level are required.
Line 901 to 916. The clone ST410 has been mainly identified in extraintestinal E. coli and is becoming increasingly significant as a high-risk strain. Research indicates a high frequency of likely transmission between species (106, 107). Whole genome sequencing of 10 isolates from both animals and humans has revealed a high degree of genetic similarity, distinguished by a low number of single nucleotide polymorphisms (SNPs). Several studies have highlighted the zoonotic potential of this clone, which exhibits multidrug resistance and poses clinical concerns for wildlife, humans, domestic animals, and the environment (107, 108). The ST410 clone has been reported in several countries, including China, Italy, Denmark, and Ghana (Liu et al., 2015; Piazza et al., 2018; Roer et al., 2018; Mahazu et al., 2021). Strains of ST410 have been described to carry the blaOXA-181 gene, an OXA-48-type carbapenemase that confers resistance to penicillins and carbapenems. The blaOXA-181 gene is believed to have originated from Shewanella xiamenensis, which is an environmental bacterium (112). In addition, there are reports of its identification from a natural water source in Singapore with resistance to carbapenems, as well as a similarity with isolates in Thailand. Furthermore, the IS26 insertion sequence has been mentioned as a mediator for acquiring resistance through the blaNDM-5 gene (113).

Reviewer 2 Report
Comments and Suggestions for Authors
Minor comments:
Line 73-74, 95-97: How do the type 1 fimbrial adhesins, such as FimH, contribute to the colonization and persistence of Uropathogenic Escherichia coli (UPEC) in the urinary tract?
Line 132-135: What is the role of extracellular assembly mechanisms, like the nucleation-precipitation pathway, in promoting biofilm integrity and resistance in UPEC?
Line 113-115, 127-128: How do genetic variations, such as those in the papG alleles, influence UPEC's adaptability to human hosts?
Line 67-68, 394: What are the functional implications of UPEC’s intracellular reservoirs and their role in recurrent urinary tract infections (UTIs)?
Line 151-154, 156-157: How do different immune defense mechanisms, such as uroplakin and Tamm-Horsfall proteins, interact with UPEC adhesins during infection?
Line 178-181, 184-187: What are the potential benefits and limitations of current UTI vaccines, such as Uromune and Urovaxom, in managing antibiotic-resistant UPEC strains?
Line 219-227, 233-237: How does the genetic diversity in UPEC, including clonal lineages like ST131 and ST1193, contribute to their virulence and antibiotic resistance profiles?
Line 362-364: What therapeutic strategies could target the adhesins (FimH, PapG, and CsgA) to inhibit the initial colonization step of UPEC in the urinary tract?
Comments on the Quality of English LanguageMinor comments
Author Response
REVIEWER 2
Comments and Suggestions for Authors
Minor comments:
- Line 73-74, 95-97: How do the type 1 fimbrial adhesins, such as FimH, contribute to the colonization and persistence of Uropathogenic Escherichia coli (UPEC) in the urinary tract?
R/We are grateful for the comments from reviewer 2 to enrich the manuscript. In section 1.1, “UPEC colonization of the urinary tract,” and in section 1.2.2, “Type 1 fimbriae”, additional information has been integrated as suggested by the reviewer and now reads as follows:
Lines 125 a 166
1.1. UPEC colonization of the urinary tract
The UEc and EcO25b strains presented a high prevalence of genes that encode various fimbrial adhesins, such as FimH, PapG, EcpA, and CsgA (12). These adhesins are associated mainly with the adhesion process, mediating the intimate interaction between bacteria and host cells. Additionally, each adhesin recognizes specific receptors in the host urothelium: (i) the FimH adhesin recognizes mannosylated residues, (ii) PapG adhesin variants recognize globosides (GbO1 to GbO5) in the renal epithelium, and (iii) the CsgA adhesin binds to fibronectin and fibrinogen (18).
1.2.2 Type 1 fimbriae
The expression of type 1 fimbrial proteins is regulated by two site-specific recombinases, FimB and FimE, which promote the inversion of the fimS switch region in response to stress factors that positively or negatively stimulate the expression of this fimbrial protein (20). Receptors for the fimbrial adhesin FimH, which include mannosylated uroplakins (UPK1) and the α1β3 integrin, are rich in mannosylated residues and are widely distributed in the umbrella cells of the bladder epithelium. When FimH interacts with its receptors, it promotes the internalization of UPEC by reorganizing actin through the activation of RHO family GTPases. Once inside, UPEC can form intracellular bacterial communities (IBCs) and quiescent intracellular reservoirs (QIRs), which are precursors to bacterial persistence in the urinary tract (25). Our working group described the frequency of type 1 fimbriae in UPEC strains from complicated UTIs [92.1% (164/178)] and of clone 025b/ST131 [95.2% (120/126)] in pediatric patients (13-14).
- Line 132-135: What is the role of extracellular assembly mechanisms, like the nucleation-precipitation pathway, in promoting biofilm integrity and resistance in UPEC?
R/ We appreciate the insightful comments provided by Reviewer 2. It is important to mention that all amyloid fimbriae assemble through the nucleation-precipitation pathway, which is an extracellular assembly mechanism. The curli fimbriae are the myloid fimbriae of UPEC and other bacterial pathogens assembled by the nucleation-precipitation pathway. Curli fimbriae have been identified in several pathogens as structures responsible for biofilm formation. Biofilms are complex and organized structures that promote antibiotic and antimicrobial resistance as a bacterial protection mechanism. In section 1.2.4, “Curli fimbriae,” several modifications have been made as suggested by reviewer 2, as described below:
1.2.4. Curli fimbriae
Amyloid structures such as curli fimbriae use an extracellular assembly mechanism called the nucleation/precipitation (NP) pathway, also called the type VIII secretion system. A wide variety of microorganisms produce functional amyloid components as structural support that favors the integrity of biofilms, a mechanism that promotes the colonization of abiotic and biotic surfaces (23). The UPEC biofilm is essential for the persistence and recurrence of UTIs. Biofilms protect bacteria from harsh conditions, antimicrobial agents, and antibiotics, and they also safeguard bacteria from the host's immune response (23). These biofilms can easily form on biological surfaces and medical devices, such as urinary catheters and the uroepithelium (36).
- Line 113-115, 127-128: How do genetic variations, such as those in the papG alleles, influence UPEC's adaptability to human hosts?
R/ We appreciate your feedback on enhancing this manuscript. It is important to describe the genetic variations found in the adhesins PapGI, PapGII, and PapGIII, as well as to define the type of interaction these adhesins have with globoids, which serve as specific ligands of the renal epithelium. In this context, we have outlined the modifications suggested by the reviewer below:
Lines 168 to 229.
In the section “1.2.3. P fimbriae”
The distal part of pyelonephritis-associated fimbriae is anchored to the PapG adhesin, which interacts with glycosphingolipids consisting of Gal-α-(1,4) Gal residues. Three alleles have been described: PapGI, PapGII, and PapGII, which bind, respectively, and differentially, to the following globoside variants: GbO3, GbO4 (both abundant in human uroepithelial cells), and GbO5 (abundant in canine cells but not in human cells) (11). PapG variations in clinical strains of UPEC are an adaptation mechanism of bacteria that allows high adaptability to colonize and cause infection in humans efficiently (21). Recent studies have shown a strong association between the PapGII variant in clinical UPEC strains belonging to the B2 phylogenetic group and several medical conditions. This variant is related to the development of pyelonephritis in adult women and children, acute prostatitis in men, and cases of bacteremia (27-32). In contrast, the PapGI variant has been found to have a low prevalence in humans with various clinical syndromes (33).
Furthermore, additional references have been added to the manuscript that substantiate its content.
- Line 67-68, 394: What are the functional implications of UPEC’s intracellular reservoirs and their role in recurrent urinary tract infections (UTIs)?
R/ We appreciate your feedback and suggestions for enhancing this manuscript. Briefly, intracellular bacterial communities (IBCs) develop following the internalization of UPEC in the bladder epithelium. Once inside the cell, the bacteria multiply the exacerbated process, similar to the formation of external biofilms, where bacterial populations are protected from adverse extracellular conditions. Additionally, IBCs promote the exfoliation of epithelial cells, which helps displace the bacterial communities and exposes the transitional epithelium, allowing bacteria to internalize and form quiescent intracellular reservoirs (QIRs). Bacteria within an IBC can exit the cell by filamentation, initiating a new infection cycle. IBCs and QIRs are central to recurrent urinary tract infections (rUTIs); once internalized, UPEC can persist for days, months, or even years, leading to new episodes of rUTIs that are resistant to both antibiotic and topical treatments.
The manuscript has been revised, incorporating new infromation in section 1.1. “UPEC Colonization of the Urinary Tract,” as described below:
Lines 106 to 122. During the intracellular process, bacteria evade the immune response and the action of antibiotics, and persistence is promoted through the formation of intracellular bacterial communities (IBCs) and quiescent intracellular reservoirs (QIRs). IBCs play several critical roles in UTIs: (i) they help bacteria persist in t UTIs, (ii) they protect bacteria from antibiotics and antimicrobial treatments, (iii) they shield bacteria from the host's immune response, (iv) they promote cell exfoliation facilitating exposure of the transitional epithelium, and (v) reactivation of IBCs enables UPEC to exit through filamentation, initiating a new infectious cycle. This process leads to rUTIs and supports bacterial persistence and reactivation once the triggering stimulus has ceased (11).
- Line 151-154, 156-157: How do different immune defense mechanisms, such as uroplakin and Tamm-Horsfall proteins, interact with UPEC adhesins during infection?
R/ We appreciate the valuable comments from reviewer 2 that have enhanced this manuscript. In section 1.3, “Immunity UTIs,” we have incorporated new information as requested by the reviewer, as described below:
Lines 290 to 312.
Three defense mechanisms against bacterial pathogens are present in the bladder: (1) physical barriers, (2) expressed peptides and proteins, and (3) immune cells (26). The urothelium in the bladder is the first physical barrier and is made up of three to six layers of transitional epithelial cells. Umbrella cells constitute the first layer of cells in the lumen of the bladder and are coated with the protein uroplakin, which functions as a physical barrier to liquids, toxins, and microorganisms, that do not express receptor proteins for UPK and can adhere to the urothelium (27). The umbrella cells are covered with mucus, composed of proteoglycans and glycosaminoglycans, and thus create a natural impermeable barrier (27). The Tamm-Horsfall protein is a highly glycosylated protein, has many cysteines (disulfide bridges), is the most abundant glycoprotein in urine, and it is produced exclusively in the tubular epithelial cells of the ascending limb of Henle (28).
The Tamm-Horsfall protein competes with uroplakin for interaction with FimH. This competition partially prevents the adherence of uropathogens that have fimbrial proteins, including FimH, which recognize and bind to mannosylated uroplakins. The structure of the Tamm-Horsfall protein is rich in mannose and contains disaccharides that bind to type I fimbriae, effectively competing with the mannose receptors on BECs. This action reduces the adhesion and colonization of UPEC in the bladder, facilitating their elimination through the urinary stream. Furthermore, the Tamm-Horsfall protein helps prevent excessive inflammation during bladder infection by inhibiting chemotaxis and reactive oxygen species (ROS) production by binding to the Ig-like lectin-9 (Siglec-9) receptor of neutrophils (29: Li et al. 2022). Secretory IgA (sIgA) is also produced in the urinary tract; sIgA is produced by the plasma cells of the lamina propria and is subsequently secreted on the surface of the mucosa, in the intestine, and other organs (30).
- Line 178-181, 184-187: What are the potential benefits and limitations of current UTI vaccines, such as Uromune and Urovaxom, in managing antibiotic-resistant UPEC strains?
R/ We appreciate the feedback provided by Reviewer 2, which has enhanced the manuscript. Additional information has been added to Section 1.3.1 “Vaccines,” as the reviewer requested, as detailed below:
Lines 371 to 380. Briefly, commercial vaccines for UTIs have demonstrated promising protection results; however, there are also reports of adverse outcomes with no noticeable improvement in patients (57). A key advantage of these vaccines is their ability to reduce the number of UTI episodes, shorten the duration of antibiotic treatment, and help prevent the development of MDR strains that are tolerant to antibiotics. The use of vaccines in UTIs caused by MDR bacteria has been controversial, although more benefits have been obtained in reducing symptoms (56). It is important to emphasize that their use for the general population is difficult due to the high costs and the fact that they do not provide 100% protection. Unfortunately, the use of commercial vaccines in pediatric patients has not been accepted to date.
- Line 219-227, 233-237: How does the genetic diversity in UPEC, including clonal lineages like ST131 and ST1193, contribute to their virulence and antibiotic resistance profiles?
How does genetic diversity in UPEC, including clonal lineages like ST131 and ST1193, contribute to their virulence and antibiotic resistance profiles?
R/ The reviewer's comment is very accurate. It is important to note that UPEC exhibits high genetic diversity. A study on UPEC strains recovered from children with complicated urinary tract infections (cUTI) in Mexico, which used whole-genome macrorestriction patterns, found that it was not possible to identify PFGE pulsotypes that could aid in differentiating hospital strains (17). In this context, the identification of types sequences through Multilocus Sequence Typing (MLST) has enabled the grouping of UPEC strains into closely related sequence types (STs) and clonal complexes (CCs), which share important virulence and resistance traits. For example, strains from clonal lineage ST131 are associated with phylogenetic group B2 associated with upper UTIs and express Extended-Spectrum Beta-Lactamases (ESBL) such as CTX-15.
In contrast, clonal lineage ST1193 has been characterized by different CCs and STs, which may be related to different geographic regions worldwide. Identifying these lineages can help infer specific strains' virulence and resistance profiles. High-risk clones have been related to characterizing virulence and antibiotic-resistance genes. Current genomic studies have allowed the identification of several UPEC lineages, posing a significant challenge for global public health. In particular, ST1193 is an emerging lineage that has been identified in UTIs and systemic infections in humans, domestic animals, and the environment. On the other hand, lineage ST131 has diversified, showing resistance to fluoroquinolones and to the presence of genes linked to cephalosporin resistance (blaCTX-M-15).
Considering the pertinent questions raised by Reviewer 2, the key characteristics of the ST131 and ST1193 lineages are described below.
Lines 698 to 704. Interestingly, ST1193 strains have high activity in promoting adherence to the human bladder cell line HTB-5 and non-biofilm formation (67) (Figure 1). E. coli strains ST131 and ST1193 pose a significant risk to human health due to their resistance to multiple antibiotics and widespread global distribution. Both strains have developed this resistance through mutations in the QRDR region, which is responsible for quinolone resistance. Additionally, they can carry IncF plasmids and virulence factors contributing to their prevalence among E. coli isolates (63, 64).
- Line 362-364: What therapeutic strategies could target the adhesins (FimH, PapG, and CsgA) to inhibit the initial colonization step of UPEC in the urinary tract?
R/ We appreciate the comments from Reviewer 2, which have enhanced this manuscript. The information requested by the reviewer is detailed below.
Lines 919 to 938. The pathogenic cycle of UPEC in the urinary tract begins with the expression of various colonization factors, which are essential as a first step for pathogenicity. The adhesins CsgA, FimH, and PapG are commonly found across different UPEC lineages and play a crucial role in the initial colonization process of the urinary tract. This step is fundamental to the pathogenic cycle of bacteria. Therefore, a cutting-edge therapeutic strategy will focus on creating a chimeric fusion protein that combines these three adhesins. This fusion protein could serve as a specific vaccine against UPEC UTIs by generating protective memory antibodies that reduce or block the initial adherence to the bladder epithelium (Figure 2).

Reviewer 3 Report
Comments and Suggestions for Authors
In general, the work is well written and understandable. However, there are points that need to be resolved:
In the introduction, check the reference on lines 41 and 42.
The presentation of the contextualisation of the problem could be better highlighted, starting with global data (if identified) and moving on to examples such as Mexico.
Figure 1 is very well constructed, but some components are in low resolution, such as the one relating to ST131.
Author Response
REVIEWER 3
Comments and Suggestions for Authors
In general, the work is well written and understandable. However, there are points that need to be resolved:
- In the introduction, check the reference on lines 41 and 42.
R/ We appreciate the comment on reviewing these references. The manuscript has included a link where relevant information on UTIs in Mexico is addressed as is described below.
Line 43 to 44. The following link has been included in the maniscript “ https://epidemiologia.salud.gob.mx/anuario/2023/principales/nacional/grupo_edad.pdf”.
- The presentation of the contextualisation of the problem could be better highlighted, starting with global data (if identified) and moving on to examples such as Mexico.
R/We appreciate the feedback from reviewer 3, and it has been very valuable for him to contribute to improving the manuscript. The information integrated into the manuscript is described below:
Line 44 to 64. Recent data from the "Global Burden of Disease Study 2019" revealed the incidence, mortality, and disability-adjusted life years (DALYs) associated with UTIs across 204 countries from 1990 to 2019 (3). In addition, these studies were conducted at different sociodemographic levels, national, regional, sex, and age levels. The analysis of these studies showed that the absolute number of UTIs cases increased by 60.40%, from 252.25 million (95% CI: 223.31-279.3) in 1990 to 404.61 million (95% CI: 359.43-446.55) in 2019 (3).
- Figure 1 is very well constructed, but some components are in low resolution, such as the one relating to ST131.
We thank the reviewer for their comment. As recommended, the image has been revised to a higher resolution of 600 dpi to ensure quality.
In the caption for Figure 1, additional relevant information has been included to enhance the description. The information that has been integrated is described below:
“The purple boxes highlight the changes in genes associated with fluoroquinolone resistance in the ST131 lineage, specifically the S83L and D87N mutations (64), as well as the parC gene mutations S80I and E84N (65). For the ST405 lineage, the gyrA gene exhibits S83L and D87N substitutions, while the parC gene shows an S80I substitution, along with a substitution in the parE gene (S458A) (98). Finally, for the ST1193 lineage, the gyrA gene has S83L and D87N mutations, and the parC gene has the S80I mutation (63).”

Round 2
Reviewer 1 Report
Comments and Suggestions for Authors I have reviewed this manuscript previously and it has significantly improved. However, there is still room for further improvement which is required. In particular, the authors need to carefully read their manuscript and revise it according to correct use of the ‘terminus technicus’ and context in a multitude of instances (a list of issues, not comprehensive, follows below). Also, in general, a more critical and differentiated evaluation of the literature would be appreciated. In addition, another still major issue is the definition of virulence factors and consequently following up on definitions. While this part of the manuscript has significantly improved, in its description, there are still some open questions. For example, the adhesin FimH inhibits sequence and functional polymorphism with alleles to possess different affinities to different mannose containing structures. Also, as the authors mention themselves in the manuscript, only certain alleles of the PapG adhesin are associated with severe disease. However, in other instances, the authors describe indifferently the presence of PapG adhesins, or even the presence of PapA which gives limited epidemiological information. In the case of curli, those amyloid fimbriae, equally as the type 1 fimbriae, but in contrast to the P-fimbriae, are present on almost all E. coli strains not only pathogens. So how do these factors specifically contribute to UTI? This is not addressed in the manuscript at all. Specific comments: Abstract: high genetic load of virulence, multiple and diverse virulence factors present resistance to structurally diverse antibiotics high frequency of extended-spectrum beta-lactamases l. 52 and elsewhere: reveled? Do you mean revealed l. 57: Are there any studies that indicate that unrecognized UTI has more severe follow-up consequences than for adults? l.65: Define MDR and XDR upon first time use. l. 77-82: Several of these statements need a reference. l. 94: UPEC colonization… of the urinary tract? l. 95: adhesion, where? l. 96: What are active host cells? l.97: What is the ‘intracellular process’? l. 159: Are these numbers, characteristics of the type 1 fimbriae substantially different from commensal and other clinical isolates of E. coli? l. 184: EcO25b collection, can this be related to the description of the sequence types that follows later in the manuscript. l.194: There is also type 1 fimbriae dependent biofilm formation. What is known which type of biofilm is most relevantrr in the urinary tract? l. 214: phylogenetic group B2, mentioned here first time. How is this classification related to UTI isolates? l. 322: How is CC related to ST, perhaps use only one nomenclature, and if using alternatives one, related to the other. l. 340: clones are high risk factors? Do you mean strains of those clones have/had a high likelihood to readily acquire mutations or genetic elements that mediate antimicrobial resistance? l. 372:Still not clear relationship betweenO25b, O16 and ST131? l. 387: Is related to phylogroup B2? Do you mean the strains of this clone/this clone belongs to phylogroup B2. Are there any exceptions? l. 426, 428, 499 and elsewhere: E. coli strains ST131 and ST1193 These are not strains, but clones l. 499: which species? Figure 1: Still different types and level of information is indicated for the different ST types. Should be harmonized. Figure 2: To this reviewer’s knowledge, E. coli has peritrichous flagella.Author Response
COMMENTS REVIEWER 1
- I have reviewed this manuscript previously and it has significantly improved. However, there is still room for further improvement which is required.
R/ We thank Reviewer 1 for their time and input on this manuscript. With the new observations addressed, we aim to enhance the understanding of the manuscript.
- In particular, the authors need to carefully read their manuscript and revise it according to correct use of the ‘terminus technicus’ and context in a multitude of instances (a list of issues, not comprehensive, follows below).
R/ We have thoroughly reviewed the manuscript to identify and correct any technical inconsistencies that might impede understanding. Thank you very much for all your feedback.
- Also, in general, a more critical and differentiated evaluation of the literature would be appreciated. In addition, another still major issue is the definition of virulence factors and consequently following up on definitions. While this part of the manuscript has significantly improved, in its description, there are still some open questions.
R/ We appreciate your feedback, and the manuscript has been thoroughly reviewed. Additionally, we have incorporated new information as requested by Reviewer 1.
- For example, the adhesin FimH inhibits sequence and functional polymorphism with alleles to possess different affinities to different mannose containing structures.
R/ We appreciate the comments from Reviewer 1, which have added valuable insights to this work. Reviewer 1 is correct in noting that variations in FimH alleles exhibit different affinities for mannose-containing structures.
We have included new information regarding the functional and sequence polymorphisms associated with the FimH adhesin, as detailed below:
Line 244 to 266. The FimH adhesin contains two domains: the pilin domain, which interacts with the FimG protein, forming a flexible junction, and the lectin domain, which forms a binding pocket for a receptor that recognizes mannosylated residues (23). The FimH adhesin exhibits genetic variability within the fimH gene, leading to the identification of several polymorphisms that give rise to different versions of the FimH protein. These polymorphisms can induce functional changes, impacting the ability of bacteria to adhere to various surfaces and tissues within their host (30, 31). A study involving E. coli strains collected from the intestinal mucosa of children with inflammatory bowel disease revealed that the FimH protein undergoes modifications in its amino acid sequence. These data suggest that commensal strains can adapt to changes in their microenvironment (32).
Although the exact number of polymorphisms in fimH gen has not been undetermined, the high frequency of structural mutations in the fimbrial adhesins of extraintestinal pathogenic E. coli (ExPEC) indicates significant diversity in their sequences. This genetic variability of FimH enhances the adaptability of E. coli to different ecological niches and conditions within the host, thereby increasing its potential to cause infections (33). An important polymorphism in UPEC strains of the ST131 lineage is FimH30. This polymorphism features an R166H mutation that weakens the interactions between the FimH domains. As a result, this change promotes stronger interactions with mannose and allows for the formation of high-affinity relaxed conformations. The expression of the FimH30 polymorphism in an isogenic E. coli strain of the ST131 lineage enhances adherence and invasion to human cells, and facilitates the formation of highly structured biofilms compared to other variants (34). The expression of type 1 fimbrial proteins is regulated by two site-specific recombinases, FimB and FimE, which promote the inversion of the fimS switch region in response to stress factors that positively or negatively stimulate the expression of this fimbrial protein (24).
- Also, as the authors mention themselves in the manuscript, only certain alleles of the PapG adhesin are associated with severe disease. However, in other instances, the authors describe indifferently the presence of PapG adhesins, or even the presence of PapA which gives limited epidemiological information.
R/ We appreciate the feedback provided by Reviewer 1. Several comments have been made regarding the reviewer's suggestions.
Lines 165 to 169. A brief explanation has been provided in response to the comment from the reviewer:
Recently, the prevalence of pap alleles was reported to be 34.3% (61/178) for papGII and 1.7% (3/178) for papGIII in a collection of UEc strains…
R/ In their study, Luna-Pineda et al. (2018) reported the amplification of specific genes, papGI, papGII, and papGIII, through multiplex PCR to perform PapG allele typing. This study was conducted on a collection of 178 urinary E. coli strains (UEc strains) obtained from children with complicated urinary tract infections (cUTIs) at the Federico Gómez Children's Hospital in Mexico City, a third-level care facility. The objective of the study was to determine the prevalence percentages of PapG alleles in the pediatric population.
Lines 248 to 252. A brief explanation has been provided in response to the comment from the reviewer:
…
Interestingly, a higher prevalence was observed from an EcO25b collection; 80.9% (102/126) amplified the papGII gene, and 2.4% (3/126) amplified the papGIII gene. In contrast, the papGI allele has not been identified in pediatric clinical strains of UPEC (16-17).
R/ The study conducted by Contreras-Alvarado et al. (2021) aimed to determine the prevalence of PapG alleles in a collection of UPEC serogroup O25b (EcO25b) strains obtained from pediatric patients hospitalized with cUTI at HIMFG. The results indicated a higher prevalence of the papGII allele in strains belonging to phylogroup B2 and serogroup O25b. Additionally, the major subunit of the P fimbrial assembly, known as PapA, was utilized to identify the presence of P fimbriae in strains that tested negative for the three PapG alleles.
A brief explanation has been provided in response to the comment from the reviewer:
Lines 317 to 322. Recently, the prevalence of pap alleles was reported to be 34.3% (61/178) for papGII and 1.7% (3/178) for papGIII in a collection of UEc strains (22). In-terestingly, a higher prevalence was observed in the EcO25b collection, associated with ST131 and phylogroup B2; 80.9% (102/126) amplified the papGII gene, and 2.4% (3/126) amplified the papGIII gene. In contrast, the papGI allele has not been identified in pe-diatric clinical strains of UPEC (23).
- In the case, those amyloid fimbriae, equally as the type 1 fimbriae, but in contrast to the P-fimbriae, are present on almost all colistrains not only pathogens. So how do these factors specifically contribute to UTI? This is not addressed in the manuscript at all.
R/ We appreciate the feedback from Reviewer 1, which will undoubtedly help improve the content of this manuscript.
Curli is an amyloid fimbria found in various strains of E. coli, including both pathogenic and commensal strains. Additionally, curli has been identified in other strains within the Enterobacteriaceae family, such as Klebsiella sp and Salmonella sp. Despite its widespread frequency, curli fimbriae have shown significant involvement in UPEC strains in the colonization process. The relevant information has been included in section "1.2.3 Curli fimbriae" (Line 179).
R/ Thank you very much to Reviewer 1 for your observation. New information has been added to the manuscript, which now reads as follows:
Lines 332 to 347. These biofilms can easily form on biological surfaces and medical devices, such as urinary catheters and the uroepithelium (36). Curli is an amyloid fimbria present in most strains of E. coli and contributes as a structure that promotes the formation in both pathogenic and commensal strains. Research has shown that curli-dependent biofilms are particularly important in UTIs associated with medical devices, such as catheters and urinary probes (48, 49). It is crucial to note that many pathogens in the Enterobacteriaceae family, including opportunistic and commensal pathogens, produce curli. Our research group has demonstrated that this fimbria acts as an accessory protein during urinary tract colonization in a murine model using C57BL/6 mice. Our results indicate that infection with a curli-producing strain can lead to significant damage to the bladder and kidneys; however, this damage is notably reduced in strains with mutations in the csgA gene (50). Curli fimbriae are made up mainly of the CsgA protein, a fine structure with aggregation and adherence properties (37).
Specific comments:
- Abstract: high genetic load of virulence, multiple and diverse virulence factors present resistance to structurally diverse antibiotics high frequency of extended-spectrum beta-lactamases…
R/ No studies currently confirm that various virulence factors contribute to antibiotic resistance. However, it is important to note that these factors play a significant role in pathogenesis, damage to the urinary epithelium, and the persistence of UPEC. The abstract has been modified as suggested by reviewer 1 and is now described below:
Lines 23 to 23. Abstract: Urinary tract infections (UTIs) are a leading cause of illness in children and adults of all ages, with uropathogenic Escherichia coli (UPEC) being the primary agent responsible.
Lines 30 to 32. Significant genetic diversity exists among UPEC strains, and ST131 represents one of the key lineages. This lineage has a high content of virulence genes, multiple mechanisms of antibiotic resistance, and a high frequency of extended-spectrum β-lactamases (ESBLs).
- 52 and elsewhere: reveled? Do you mean revealed
R/ Thank you very much to reviewer 1 for your observation.
Line 48. The word "reveled" has been changed to "showed."
- 57: Are there any studies that indicate that unrecognized UTI has more severe follow-up consequences than for adults?
R/ The manuscript highlights the characteristics of UTIs in infants under 2 years of age, particularly because there is limited information about symptoms in this age group. In contrast, diagnosing UTIs in adults is generally more straightforward, as there is a more comprehensive understanding of the signs and symptoms. The manuscript also includes new information, as outlined below:
Line 71 to 96. The guidelines for diagnosing UTIs in infants emphasize on preventing renal scarring. Research has highlighted the long-term consequences of renal scarring, which may include reduced kidney function, hypertension, and even end-stage renal disease (7, 8).
It is important to note that after a comprehensive review, no articles were found that describe kidney damage resulting from a failed diagnosis of a UTI in adults.
Line 71 to 96. Timely diagnosis of UTIs in both children and adults should always be a priority. This approach allows for the identification of the pathogen, enabling prompt treatment and eradication. However, there are existing recommendations concerning the diagnosis and treatment of UTIs in pediatric patients. Recent studies have indicated that the urinary tract is a common source of infections in children of all ages (4). These infections come with risk factors for potential complications following UTIs, including congenital anomalies of the kidney and urinary tract, as well as vesicointestinal dysfunction. It is crucial to consider a diagnosis of UTI in every child who presents with a fever without an apparent source of infection. Several factors must be taken into account for appropriate management, such as distinguishing between a lUTIs and uUTIs, selecting the appropriate age-specific urine collection method, assessing risk factors, obtaining a positive urine culture based on the Kass-Sandford criterion, accurately identifying the uropathogen, and applying suitable treatment based on clinical observations and laboratory findings. It is important to mention that new information has been added based on the suggestion from Reviewer 1. The newly incorporated information in the manuscript is presented below.
Line 128 to 134. In conclusion, several important factors must be considered to effectively manage pediatric patients with UTIs. These factors include differentiating between lUTIs and uUTIs, using age-appropriate methods for urine collection, identifying potential risk factors, obtaining a positive urine culture in accordance with the Kass-Sandford criteria, accurately identifying the uropathogen, and applying appropriate treatment based on clinical and laboratory guidelines (4).
- 65: Define MDR and XDR upon first time use.
R/ We appreciate Reviewer 1's comment. The manuscript now clearly describes MDR and XDR, including their meanings.
Line 135 to 139. ….. multidrug-resistant (MDR) and extensively drug-resistant (XDR). An MDR strain is defined as one that is resistant to at least one antibiotic from three or more different classes of antimicrobial agents. In contrast, an XDR strain is resistant to at least one antibiotic from all classes but no more than two classes (9).
- 77-82: Several of these statements need a reference.
R/ We appreciate the comments made by Reviewer 1, which have helped us enhance the manuscript. Following the reviewer's suggestion, we have included additional references, as outlined below:
Line 116. Flores-Mireles AL, Walker JN, Caparon M, Hultgren, S. J. 2015. Urinary tract infections: epidemiology, mechanisms of infection and treatment options. Nature reviews. Microbiology, 13(5), 269–284.
Line 118. Terlizzi ME, Gribaudo G, Maffei ME. UroPathogenic Escherichia coli (UPEC) Infections: Virulence Factors, Bladder Responses, Antibiotic, and Non-antibiotic Antimicrobial Strategies. Front Microbiol. 2017 Aug 15;8:1566.
Line 121. Ochoa, S. A., Cruz-Córdova, A., Luna-Pineda, V. M., Reyes-Grajeda, J. P., Cázares-Domínguez, V., Escalona, G., Sepúlveda-González, M. E., López-Montiel, F., Arellano-Galindo, J., López-Martínez, B., Parra-Ortega, I., Giono-Cerezo, S., Hernández-Castro, R., de la Rosa-Zamboni, D., & Xicohtencatl-Cortes, J. (2016). Multidrug- and Extensively Drug-Resistant Uropathogenic Escherichia coli Clinical Strains: Phylogenetic Groups Widely Associated with Integrons Maintain High Genetic Diversity. Frontiers in microbiology, 7, 2042.
- 94: UPEC colonization… of the urinary tract?
Thank you for your feedback. All the relevant information has been included in lines 130 to 137, as shown below.
- 95: adhesion, where?
R/ Thank very much for the observation.
Line 175. The reviewer suggested including the statement: “adhesion to the bladder epithelium."
- 96: What are active host cells?
R/ Thank very much for the suggesting.
Line 133. The phrase "active host cells" was replaced with "umbrella cells."
- 97: What is the ‘intracellular process’?
R/ We thank reviewer 1 for the comment. The phrase "intracellular process" has been replaced with "intracellular invasion process." Additional information has been included in the manuscript, as shown below.
We will provide a brief explanation of these two concepts.
The term "Intracellular processes" describes the chemical and physical reactions that occur inside a cell and are essential for the cell's survival. These processes include energy intake, stimulus reception, metabolic activities, waste energy recovery, and genetic regulatory mechanisms.
The term "intracellular invasion processes" describes how pathogens enter and multiply within host cells. This invasion often requires overcoming the host cell's plasma membrane and cytoskeleton. Strategies that facilitate intracellular invasion include the formation of moving junctions, the use of endocytic pathways, actin polymerization, and the modulation of the actin cytoskeleton.
The modifications performed in the manuscript according to review 1 are described below:
Line 174 to 181. UPEC colonization is the initial event that promotes the establishment of urinary tract infection in the host, subsequently leading to a UTI. UPEC adheres to the bladder epithelium as part of the colonization process, allowing it to evade urine clearance, interact with the “umbrella cells” of the urinary epithelium, and, in some cases, undergo an intracellular invasion process (14). UPEC can internalize and multiply within host cells during this intracellular invasion, helping it evade the immune response. This capability diminishes the effectiveness of antibiotics and promotes persistence by forming intracellular bacterial communities (IBCs) and quiescent intracellular reservoirs (QIRs).
- 159: Are these numbers, characteristics of the type 1 fimbriae substantially different from commensal and other clinical isolates of E. coli?
New information has been added to the manuscript, as detailed below:
Line 275 to 279. Our working group described the frequency of type 1 fimbriae in UPEC strains from complicated UTIs [92.1% (164/178)] and of clone 025b/ST131 [95.2% (120/126)] in pediatric patients (16-17). Our findings align with previous observations in commensal E. coli strains and other clinical samples, mainly due to the high conservation of type 1 fimbriae in this microorganism (36).
- 184: EcO25b collection, can this be related to the description of the sequence types that follows later in the manuscript.
R/ We appreciate your observation. In this collection of UPEC strains from serogroup O25b, we found an association with clonal lineage ST131, which is part of the clonal complex CC131 and is related to phylogroup B2. It is important to note that two distinct studies are mentioned in the manuscript. The first study, conducted by Luna-Pineda et al. in 2018, analyzed 178 urinary E. coli strains and identified, for the first time, strains of serotype O25b in pediatric patients. Later, in 2021, Contreras Alvarado et al. characterized the population of UPEC/O25b (EcO25b) strains using macrorestriction techniques (PFGE) and MLST. For clarification, the references for each study are indicated in the manuscript below.
Line 319. The reference by Luna-Pineda et al., 2018b was included in the manuscript.
Line 322. The reference by Contreras-Alvarado et al., 2021 was included in the manuscript.
Line 245 to 250. The sentence “a higher prevalence was observed in the EcO25b collection, associated with ST131 and phylogroup B2" was modified and included in the manuscript.
Recently, the prevalence of pap alleles was reported to be 34.3% (61/178) for papGII and 1.7% (3/178) for papGIII in a collection of UEc strains (22). Interestingly, a higher prevalence was observed in the EcO25b collection, associated with ST131 and phylogroup B2; 80.9% (102/126) amplified the papGII gene, and 2.4% (3/126) amplified the papGIII gene. In contrast, the papGI allele has not been identified in pediatric clinical strains of UPEC (23).
- 194: There is also type 1 fimbriae dependent biofilm formation. What is known which type of biofilm is most relevant in the urinary tract?
R/ We appreciate the comment that will enhance the manuscript. A brief comment and reference regarding studies of type 1 fimbria-dependent biofilms and urinary tract infections have been added. The information included in the manuscript is shown below:
Line 332 to 347. These biofilms can easily form on biological surfaces and medical devices, such as urinary catheters and the uroepithelium (36). Curli is an amyloid fimbria present in most strains of E. coli, and its expression promotes the formation of biofilms in both pathogenic and commensal strains. In addition, curli-dependent biofilms can play a significant role in urinary tract infections associated with medical devices, such as catheters and urinary probes (48, 49). While many pathogenic, opportunistic and commensal of the Enterobacteriaceae family produce curli, our research group has demonstrated that curli fimbria functions as an accessory protein that enhances urinary tract colonization in a C57BL/6 mouse model. The results indicate that infection with a curli-producing strain can cause more significant damage to the bladder and kidney than the damage caused by the same strain mutated in the csgA gene (50). Although the curli fimbria is directly involved in the formation of biofilms by UPEC strains, type 1 fimbria-dependent biofilms have also been reported on abiotic surfaces (51). These biofilms contribute to the persistence of catheter-associated urinary tract infections (CAUTI) (52) and are a crucial factor in the development of UTIs (53). Furthermore, type 1 fimbria facilitates colonization of the bladder epithelium and the IBCs, contributing to persistent and recurrent UTI infections. Curli fimbriae are made up mainly of the CsgA protein, a fine structure with aggregation and adherence properties (37).
- 214: phylogenetic group B2, mentioned here first time. How is this classification related to UTI isolates?
R/We consider this comment from Reviewer 1 to be important. New information has been incorporated into the manuscript, as described below:
Line 298 to 306. Recent studies have shown a strong association between the PapGII variant in clinical UPEC strains belonging to the B2 phylogenetic group and several medical conditions. Phylogroup B2 is recognized as a pathogenic group associated with UTIs and has been characterized by genes coding for various virulence factors, including adhesin genes, toxins, and iron acquisition systems (38, 39, 40). Strains belonging to this phylogroup are also known for their ability to form biofilms (41), which enhances their pathogenicity and increases the likelihood of severe infections. Furthermore, UPEC strains from phylogroup B2 have shown an increasing trend in acquiring multidrug resistance (42). This variant is related to the development of pyelonephritis in adult women and children, acute prostatitis in men, and cases of bacteremia (27, 28, 29, 30). Additionally, the PapGIII variant has been linked to the occurrence of cystitis in women, men, and children (31, 32). In contrast, the PapGI variant has a low prevalence in humans with various clinical syndromes (33).
- 322: How is CC related to ST, perhaps use only one nomenclature, and if using alternatives one, related to the other.
R/ The reviewer's comments are very insightful and will significantly enhance the manuscript. At this stage, we would like to offer some brief comments: A Sequence Type (ST) is a technique used to analyze microorganism samples by sequencing DNA from multiple genes.
Multilocus Sequence Typing (MLST) specifically involves sequencing short fragments of multiple "housekeeping" genes to identify and track bacterial pathogens. This technique is crucial for studying the global epidemiology of bacteria and identifying antibiotic-resistant strains. A collection of STs, which share a percentage of similarity based on a central allelic profile (genotype), can be identified and grouped to form CCs using heuristic methods such as eBURST. CCs play a vital role in identifying bacteria related to nosocomial infections, informing treatment decisions, and conducting epidemiological surveillance. The study of CCs has helped conceptualize biological species and the best and most up-to-date taxonomic classification. Given this scenario, it is not convenient to use them as a single nomenclature, in addition some STs do not originate a CC, but do belong to a CC, in this case both must be indicated. The study of CCs has contributed to understanding biological species and developing contemporary taxonomic classifications. Given this scenario, it is not convenient to use them as a single nomenclature, in addition some STs do not originate a CC, but do belong to a CC, in this case both must be indicated. To enhance the manuscript, we have thoroughly reviewed all content related to STs and CCs, indicating the information more precisely and concisely. Spratt BG. Exploring the concept of clonality in bacteria. Methods Mol Biol. 2004;266:323-52. doi: 10.1385/1-59259-763-7:323. PMID: 15148426.
The changes are reflected in lines 489, 502, 515, 519, 521, 528, 531, 555, 582, 583, 590 and 611.
- 340: clones are high risk factors? Do you mean strains of those clones have/had a high likelihood to readily acquire mutations or genetic elements that mediate antimicrobial resistance?
R/ We appreciate Reviewer 1 for their insightful observation. There is an error in the manuscript. In fact, clones are not high-risk factors. The correction was made to say “clone” instead of “factors.” The changes and comments made to the manuscript are listed below.
- clones are high risk factors?
Line 566 to 569. ST131 clones responsible for UTIs, are high risk clones for resistance to multiple antimicrobial determinants and have MDR and XDR phenotypes (82). High-risk clones are distributed globally and associated with several antimicrobial resistance determinants, which aid their transmission and persistence in hosts (83). Tn addition, ST131 clones contain the blaCTX-M-15 gene, which confers resistance to cephalosporins.
- Do you mean strains of those clones have/had a high likelihood to readily acquire mutations or genetic elements that mediate antimicrobial resistance?
R/ Yes, the importance of the different clonal lineages identified in UPEC is that they had and may continue to have a high probability of not only mutations, insertions, and horizontal transfer of mobile genetic elements that confer the characteristic of antibiotic resistance. All these characteristics can be acquired during geographic dissemination, ecological niches, environmental and/or hospital stay, stay in the host, and competition with the microbiota. In addition, high-risk clones are globally distributed clones, and their association with several determinants of antimicrobial resistance facilitates their transmission and persistence in hosts (83).
- 372:Still not clear relationship betweenO25b, O16 and ST131?
R/ We appreciate observation 1 and mention that the clones of ST131 may be associated with two main serotypes: ST131/O25b clone and ST131/O16 clone. The aim is to clarify any confusion; therefore, new information has been added, as described below:
Lines 570 to 580. The ST131 has been mainly associated to clinical UPEC strains belonging to the serogroups O25b and O16. A specific subclone of UPEC known as ST131-O16, which is part of the ST131 clonal lineage, has emerged as a variant. This subclone has been identifed in the fecal samples of both healthy individuals and patients suffering from cystitis and pyelonephritis in China (71, 89). High virulence has been observed in the ST131-O16 clones, which has been associated with subcutaneous sepsis in murine models. This virulence is linked to a high frequency of the fimH gene and the presence of specific mutations in the gyrA gene (S83L and D87N) and the parC gene (S80I), which confer fluoroquinolone resistance (89, 90). Additionally, the fimH41 allele has been correlated with increased virulence in vitro (91). In Nigeria, the ST131-O16 clone has been identified in patients diagnosed with sepsis (92).
UPEC ST131 serogroup O25b (UPEC/ST131-O25b) is the most prevalent clone identified in adult women in ICUs across China. This clone exhibits resistance to fluoroquinolones due to specific mutations in the gyrA (S83L, D87N, and A93E/G) and parC (S80I and E84N) genes, as reported by Zhong et al. (2019). Additionally, mutations in the gyrB gene have also been associated with fluoroquinolone resistance (93). Furthermore, the ST131-O25b clone has been observed to carry the fimH30, fimH41, and fimH27 alleles, according to findings by Dahbi et al. (2014) (94).
- 387: Is related to phylogroup B2? Do you mean the strains of this clone/this clone belongs to phylogroup B2. Are there any exceptions?
R/ Recent literature indicates that only one association of clone ST1193 with phylogroup B2, serogroup O75, and CC14 has been described. However, there is no evidence to suggest that any phylogenetic group other than B2 is associated with ST1193 clones.
Line 481 to 483. The following reference has been included in the following statement. ST1193 belongs to clonal complex 14 (CC14), which is an O75 clonal group; is a nonlactose fermenter; is related to phylogenetic group B2; and is resistant to quinolones (105, 109-110), maintaining a correlation with ST131 (86-88).
- 426, 428, 499 and elsewhere: E. coli strains ST131 and ST1193 These are not strains, but clones
R/ We appreciate the feedback from Reviewer 1. The working group referred to "ST" as a characteristic of clinical isolates, which is why the term "strain" was initially used in the manuscript. However, in light of the clonal diversity discussed in the document, we have decided to adopt the reviewer’s suggestion and replace "strain" with "clones." The manuscript has been thoroughly revised, and these changes have been consistently applied throughout the document using the terms "clones ST131" and "clones ST1193."
- 499: which species? Figure 1: Still different types and level of information is indicated for the different ST types. Should be harmonized. Figure 2: To this reviewer’s knowledge, E. coli has peritrichous flagella.
Thank you for your feedback on the first revision. The changes we made are outlined below.
Line 881 to 884. The clone ST410 has been predominantly identified in extraintestinal E. coli and is increasingly recognized as a high-risk clone. Research demonstrates a significant frequency of interspecies transmission (humans, poultry, companion animals, and sewage) in Germany.
Figure 1: Still different types and level of information is indicated for the different ST types. Should be harmonized.
Figure 1: Different types and levels of information for ST should be harmonized.
R/ We appreciate Reviewer 1's feedback. Figure 1 has been modified to ensure consistency across all STs. The changes made to the figure are detailed below.
- Figure 1 includes orange boxes representing clade diversification for each ST, and here, this information is not available; it is labeled as "Without clade diversification."
- In ST8196, details regarding C2/H30-Rx, serotype, and substitutions that confer resistance to fluoroquinolones are provided.
- In ST101, information about phylogenetic group B1 has been added.
- The figure also depicts the bacteria along with their peritrichous flagella.
Figure 2: To this reviewer’s knowledge, E. coli has peritrichous flagella.
We appreciate the comment from reviewer one on improving the simulation to be as close to reality as possible. Figure 2 shows an E. coli with peritrichous flagella.
Round 3
Reviewer 1 Report
Comments and Suggestions for Authors
Please read the manuscript one more time carefully.
For example, there are repetitions (see line 72-83).
Author Response
REVIEWER COMMENTS 1
Please read the manuscript one more time carefully.
We appreciate the comments provided by the reviewer. The authors have read, reviewed, and modified several sentences in the manuscript. The changes made are outlined below.
Page 2, line 79 to 83. The following information “These factors include differentiating between lUTIs and uUTIs, using age-appropriate methods for urine collection, identifying potential risk factors, obtaining a positive urine culture in accordance with the Kass–Sandford criteria, accurately identifying the uropathogen, and applying appropriate treatment on the basis of clinical and laboratory guidelines (4)” was deleted in the manuscript due to repeated information. Thank you for this observation.
Page 4, lines 104 to 107. The following paragraph has been modified and now reads as follows: “Our working group reported that one of the main problems with UTIs in the pediatric population is the presence of MDR and XDR clinical strains of UPEC and the production of extended-spectrum beta-lactamase (ESBL) (16)”.
Page 4, lines 116 and 116. The following paragraph has been modified and now reads as follows: “Colonization is the initial event that promotes the establishment of UPEC in the UT of the host, subsequently leading to a UTI”.
Page 5, lines 219 and 220. The following sentence was removed from the manuscript “Recently, the prevalence of pap alleles was reported to be 34.3% (61/178) for papGII and 1.7% (3/178) for papGIII in a collection of UEc strains” for having repeated information.
Page 7, lines 337 to 338. The following statement was modified for clarity “However, side effects, such as neck rash and fever, and gastrointestinal problems, such as diarrhea, decreased appetite, and nausea, have been reported (77) (Table 1)” y ahora se lee de la siguiente manera “However, side effects, such as burning sensation, low grade fever, nauseas, vaginal rash, vaginal bleeding, have been reported (77)”.
Page 12, line 501 to 505. The following statement “UPECST131 clones and ST1193 clones pose a significant risk to human health because of their resistance to multiple antibiotics and widespread global distribution” it was modified and now reads as follows “Interestingly, the clone ST1193 has high activity in promoting adherence to the human bladder cell line HTB-5 and nonbiofilm formation (127) (Figure 1). ST131 and ST1193 clones of UPEC pose a significant risk to human health because of their resistance to multiple antibiotics and widespread global distribution”.
Page 15, lines 649 to 652. The following information was integrated into the manuscript “Jesús David García-García, and Laura M. Contreras-Alvarado also received support from a doctoral scholarship from SECIHTI (Secretaría de Ciencia, Humanidades, Tecnología e Innovación) of Mexico with CVU 958410 and 963336, respectively”.
Note
The manuscript has been uploaded to the microorganism system as a Word document with change control, along with a final version without change control in PDF format.
